# Assessment of carbonaceous aerosols in Shanghai, China, Part 1: Long-term evolution, seasonal variations and meteorological effects

Yunhua Chang[1, 2, *], Congrui Deng[3], Fang Cao[1, 2], Chang Cao[1, 2], Zhong Zou[4], Shoudong Liu[1, 2], Xuhui Lee[1, 2, 5], Jun Li[6], Gan Zhang[6], and Yanlin Zhang[1, 2, *]

[1]Yale-NUIST Center on Atmospheric Environment, International Joint Laboratory on Climate and Environment Change (ILCEC), Nanjing University of Information Science & Technology, Nanjing 210044, China

[2]Key Laboratory of Meteorological Disaster, Ministry of Education (KLME)/ Collaborative Innovation Center on Forecast and Evaluation of Meteorological Disasters (CIC-FEMD), Nanjing University of Information Science & Technology, Nanjing 210044, China

[3]Shanghai Key Laboratory of Atmospheric Particle Pollution and Prevention (LAP[3]), Department of Environmental Science and Engineering, Fudan University, Shanghai 200433, China

[4]Pudong New Area Environmental Monitoring Station, Shanghai 200135, China

[5]School of Forestry and Environmental Studies, Yale University, New Haven, Connecticut 06511, USA

[6]State Key Laboratory of Organic Geochemistry, Guangzhou Institute of Geochemistry, Chinese Academy of Sciences, Guangzhou 510640, China

*Correspondence to*: Yanlin Zhang (dryanlinzhang@outlook.com or yanlinzhang@nuist.edu.cn) and Yunhua Chang (changy13@fudan.edu.cn or changy13@nuist.edu.cn)

**Abstract** Carbonaceous aerosols are major chemical components of fine particulate matter ($PM_{2.5}$) with major impacts on air quality, climate change, and human health. Gateway to fast-rising China and home of over twenty million people, Shanghai throbs as the nation's largest mega city and the biggest industrial hub. From July 2010 to December 2014, hourly mass concentrations of ambient organic carbon (OC) and elemental carbon (EC) in the $PM_{2.5}$ fraction were quasi-continuously measured in Shanghai's urban center. The annual OC and EC concentrations (mean ± 1 σ) in 2013 (8.9±6.2 and 2.6±2.1 μg m$^{-3}$, n=5547) and 2014 (7.8±4.6 and 2.1±1.6 μg m$^{-3}$, n=6914) were higher than that of 2011 (6.3±4.2 and 2.4±1.8 μg m$^{-3}$, n=8039) and 2012 (5.7±3.8 and 2.0±1.6 μg m$^{-3}$, n=4459). We integrated the results from historical field measurements (1999-2012) and satellite observations (2003-2013), concluding that carbonaceous aerosol pollution in Shanghai has gradually reduced since 2006. In terms of mmonthly variations, average OC and EC concentrations ranged from 4.0 to 15.5 and from 1.4 to 4.7 μg m$^{-3}$, accounting for 13.2-24.6% and 3.9-6.6% of the seasonal $PM_{2.5}$ mass (38.8-94.1 μg m$^{-3}$), respectively. The concentrations of EC (2.4,

2.0, 2.2, 3.0 μg m$^{-3}$ in spring, summer, fall, and winter, respectively) showed little seasonal variation (excepting winter) and weekend-weekday dependence, indicating EC are a relatively stable constitute of PM$_{2.5}$ in the Shanghai urban atmosphere. In contrast to OC (7.3, 6.8, 6.7, and 8.1 μg m$^{-3}$ in spring, summer, fall, and winter, respectively), EC showed marked diurnal cycles and correlated strongly with CO across all seasons, confirming vehicular emissions as the dominant source of EC at the targeted site. Our data also reveal that both OC and EC showed concentration gradients as a function of wind direction and wind speed, generally with higher values associated with winds from the southwest, west, and northwest. This was consistent with their higher potential as source areas, as determined by the potential source contribution function analysis. A common high potential source area, located along the middle and lower reaches of the Yangtze River instead of Northern China, was pinpointed during all seasons. These results demonstrate that the measured carbonaceous aerosols were driven by the interplay of local emissions and regional transport.

**1 Introduction**

Atmospheric carbonaceous aerosols comprise 10-70% of PM$_{2.5}$ (atmospheric particulate matter with aerodynamic diameters equal or less than 2.5 μm) mass with particularly high levels found in urban atmospheres (Turpin et al., 2000; Pöschl, 2005) . Broadly, carbonaceous aerosols have three categories, namely organic carbon (OC), elemental carbon (EC; loosely also known as black carbon (BC), and carbonate carbon (CC) (Turpin et al., 2000; Bond et al., 2013). OC is carbon associated with organic compounds either directly emitted to the atmosphere (primary OC, POC) or formed by the condensation of compounds produced via the atmospheric photochemical reaction of anthropogenic and biogenic volatile organic precursors (secondary OC, SOC) (Turpin and Huntzicker, 1991; Hallquist et al., 2009). EC is exclusively of primary origin and essentially nonvolatile, formed from incomplete combustion of fossil fuel such as automobile engines (especially diesel vehicles), coal, or biomass burning through thermal degradation of organic materials. CC, typically present in natural mineral dust and building/demolition dust, exists mainly in the coarse fraction and was found to be negligible in some studies (Sillanpaa, 2005; Chow and Watson, 2002).

Gathering evidence has shown a consistent association of the mass concentrations of carbonaceous aerosols with a range of local to global challenges including air pollution (Castro et al., 1999; Cao et al.,

2007; Huang et al., 2014), visibility impairment (Park et al., 2003; Cao et al., 2012; Volkamer et al., 2006), health damage (Pope III et al., 2002; Samet et al., 2000; Qiao et al., 2014; Lim et al., 2012), and climate change (Bond et al., 2013; Gustafsson et al., 2009; Jacobson, 2001). China is thought to be one of the leading contributors to the global burden of carbonaceous aerosols emissions (Cao et al., 2006; Klimont et al., 2009; Fu et al., 2012; Li et al., 2009; Cui et al., 2015), and this can be largely explained by its fast urbanization rate coupled with rapid industrial development in its economically developed regions (Fang et al., 2016; Zhao et al., 2015; Lin et al., 2014), such as Beijing-Tianjin-Hebei (BTH), the Yangtze River Delta (YRD; cities such as Shanghai, Nanjing and Hangzhou), and the Pearl River Delta (PRD; cities like Guangzhou, Shenzhen and Hong Kong). Estimated from the "bottom-up" methodology, the nation's vehicular emissions of BC and OC increased significantly from 47.1 Tg and 74.4 Tg in 1999 to 177.6 Tg and 101.5 Tg in 2011 (Cui et al., 2015). Co-emitted with EC and POC from on-road traffic, NMVOC (Non-methane Volatile Organic Compounds) emissions can also be expected to experience a similar trajectory (Li et al., 2015). Nevertheless, it is worth noting that vehicle-related emissions in China lagged significantly behind its double-digit growth in annual automobile sales in the 2000s (Chang et al., 2016) due to implementation of policy to curb vehicular emissions.

Change has come to China. China's National Ambient Air Quality Standard (NAAQS) implemented regulations to control $PM_{10}$, $NO_x$ and $SO_2$ concentrations in the 1990s. Starting from 2005, China had bypassed the national goal of a 10% reduction in $SO_2$ emissions as of 2010 (achieving a 14.3% reduction), and $NO_x$ emissions also had a plan to be 10% lower than the benchmark of 2010 (Chang, 2012). The focus is now on PM2.5. In 2013, China unveiled its five-year "Ten-point air plan", a comprehensive guideline calls for nationwide improvements in air quality by 2017, aiming to cut $PM_{2.5}$ levels by 25%, 20%, 15% in the regions of BTH, YRD, and PRD, respectively (MEP, 2013). Other quantified targets for 2017 include a drop of around 20% in the energy intensity of industrial added value on 2012 levels; and a fall in the percentage of coal use in total energy consumption to 65% or less. Besides, 150 billion cubic meters of new natural gas pipeline capacity will come online in the three key regions by 2015. The plan lists 33 measures to achieve the targets, including further incentives for new-energy vehicles, fuel quality improvements, dealing with small coal furnaces and reductions in coal use in the three key regions (Zhang et al., 2017).

Previous field sampling studies have shown that carbonaceous aerosols in many Chinese cities account for 20 to 50% of $PM_{2.5}$ mass (Cao et al., 2004; Cao et al., 2005; Cao et al., 2007; Duan et al., 2005; Zhang et al., 2008b; He et al., 2001; Cao et al., 2013; Duan et al., 2007; Zheng et al., 2005; Zhang et al., 2008a). Based on daily $PM_{2.5}$ filter samples and the IMPROVE protocol, Cao et al., (2007) performed the first nationwide simultaneous measurements of OC and EC in 14 cities in China during winter and summer seasons in 2003. A one-year (2006) 24-hr sampling campaign carried out at 18 different sites (including rural, urban and remote locations) was reported in (Zhang et al., 2008b), in which they provided unique insights into the seasonal and spatial variations of carbonaceous aerosol pollution across China. More recently, radiocarbon-based source apportionment of carbonaceous aerosols has been applied in China (Zhang et al., 2014; Zhang et al., 2012; Zhang et al., 2015a; Zhang et al., 2016). Nevertheless, long-term monitoring strategies based on the analysis of aerosols sampled on filters are subject to various sampling and analytical artifacts (Arhami et al., 2006; Cheng et al., 2009; Wu et al., 2016; Turpin et al., 1994); they are labor-intensive and time consuming; moreover, they fail to capture processes governing diurnal variations of atmospheric pollutants and cannot provide precise diagnostics during pollution episodes. In this context, the Sunset Laboratory semi-continuous OC/EC analyzer is capable of providing near real-time information of artifact-free measurement of carbonaceous aerosols, and has greatly improved the understanding of the sources and transformation processes of carbonaceous aerosols. However, due to the maintenance cost and intensive calibration requirements the semi-continuous OC/EC analyzer has rarely been used for long-term monitoring in China. Besides, auxiliary instruments related to aerosol chemical components and optical properties have not previously been applied to aid OC/EC data quality check and PM source apportionment.

In response to these deficiencies and needs, an in situ atmospheric superstation has been implemented in 2010 at Shanghai urban center, allowing the chemical, physical and optical characterization of PM pollution for the largest megacity in China. As the biggest megacity in China, Shanghai is one of several cities pioneering the implementation of new policies or adopting advanced technologies to curb air pollution since 2000s. Therefore, the evolution of air pollution in Shanghai, to a large extent, exemplifies the progresses and challenges toward cleaning China's air. In this study, we describe and discuss the longest (from 10 June 2010 to 31 December 2014) on-line field measurement of carbonaceous aerosols in Shanghai, China obtained by a Sunset online OC/EC analyzer, aiming to elucidate their variation

characteristics and geographical origins, examine the effects of meteorological factors on measured carbonaceous aerosols, and assess the effectiveness of control measures taken in Shanghai. Future work will focus on the sources and formation mechanisms of carbonaceous aerosols.

## 2 Experiment

### 2.1 Site description

Covering 6340.5 km²of land area and enjoying the largest commercial and industrial hub in China (Fig. 1), Shanghai is home to a population of over 24 million people and 2.7 million vehicles (0.3 million of which are diesel engines) according to figures from 2013 (China Automotive Industry Yearbook, 2014). The emissions of $SO_2$, $NO_x$, $PM_{10}$, $PM_{2.5}$, BC, OC, NMVOC, and $NH_3$ in Shanghai in 2010 were estimated as 620.3, 468.0, 160.0, 112.0, 12.5, 10.5, 541.4, and 61.8 Mg, respectively (Huang et al., 2011). Field measurements were carried out at the Pudong Environmental Monitoring Center (PEMC; 121.5447 °E, 31.2331 °N), which is located in southwest urban Shanghai with an intensive urban road network (SI Fig. S1). There were no major industrial sources nearby. The sampler inlet was on the rooftop of a 5-floor building (~18 m above the ground). There were no buildings obstructing observations at this height and the air mass could flow smoothly through the local area. The detailed description of the sampling site is given elsewhere (Chang et al., 2016).

### 2.2 Field measurements

Hourly ambient OC and EC concentrations were recorded from 10 June 2010 to 31 December 2014 through a Sunset Laboratory semi-continuous OC/EC analyzer (RT-4 model, Sunset Lab. Inc., USA). The analyzer was based on the NIOSH method 5040 and employs the thermal-optical transmittance (TOT) protocol for pyrolysis correction (Turpin et al., 2000; NIOSH, 1996). Ambient air was input with a flow rate of 8 LPM through a $PM_{2.5}$ sharp-cup cyclone to analyze OC and EC concentrations. To remove semi-volatile organic vapors that could potentially be collected onto the quartz filter media, the sampled aerosols were passed through a multichannel, parallel plate denuder with a carbon impregnated filter (CIF) and were collected on a quartz fiber filter. Significant buildup of refractory substances on the filter can occur and promote the catalytic oxidation of carbon on the filter prior to oxygen injection, thereby

affecting the accuracy of the analyzer. Therefore, the quartz fiber filter was replaced every 3-5 days by checking the laser correction factor.

Following filtration, the sampled aerosols were iteratively heated at four increasing temperature steps to vaporize organic compounds. Immediately after, the compounds were catalytically oxidized to $CO_2$ gas

over a bed of $MnO_2$ in the oxidizing oven. The $CO_2$ gas was swept out of the oxidizing oven under a stream of high purity (99.999% higher) helium atmosphere and measured by a self-contained non-dispersive infrared detector (NDIR). Following OC oxidation, EC was oxidized to $CO_2$ when the sample oven temperature is stepped up to 850°C. At the end of every analysis, the semi-continuous OC/EC analyzer was automatically calibrated by injecting a standard $CH_4$ mixture (5% $CH_4$ in He). Calibration

data, obtained from the known carbon concentration in the loop, were incorporated into every measurement to calculate the analytical results. The NDIR response to the known carbon concentration, as $CH_4$, was obtained to determine the NDIR response factor. The precision and the detection limits of the TOT instrument were determined by the variability of the EC and OC concentrations on exposed filters and on filter blanks, respectively. The precision was found to be satisfactory with a relative

standard deviation below 5%. The level of EC on the filed blanks was negligible, whereas the OC concentration ranged from 0.1-1.0 μg m$^{-3}$. For a time-resolution of 1 hr. (45 min collection) in this study, the detection limit of the semi-continuous OC/EC analyzer was 0.2 μg m$^{-3}$ for OC and 0.04 μg m$^{-3}$ for EC.

Concentrations of BC were continuously measured using an Aethalometer (Model AE-31, Magee

Scientific Company, USA) which has seven wavelengths (370, 470, 520, 590, 660, 880 and 950 nm). Measurement at 880 nm wavelength is considered as the standard channel to determine BC concentrations because absorption of radiation of other aerosols (e.g. organic aerosols) are negligible at 880 nm (Arnott et al., 2005), and thus was used in the present study. The mass concentrations of $PM_{2.5}$ were measured using a Thermo Fisher Scientific TEOM 1405-D since January 2013. Data on hourly

concentrations of CO (Thermo 48i) and $NO_2$ (Thermo 42i), daily $SO_2$ concentrations (Thermo 43i) at Pudong site (2000-2015), annual average $SO_2$ concentrations (2000-2014) and emissions (2000-2013) in Shanghai were provided by Pudong Environmental Monitoring Center. The routine QA/QC (quality assurance/quality control) procedures, including the daily zero/standard calibration, span and range check, station environmental control, and staff certification, were followed the Technical Guideline of

Automatic Stations of Ambient Air Quality in Shanghai based on the national specification HJ/T193–2005, which was modified from the technical guidance established by the USEPA. The multi-point calibrations were weekly applied upon initial installation of the instruments and the two-point calibrations were applied on a daily basis. Meteorological data, including ambient temperature ($T$), relative humidity (RH), wind direction (WD) and wind speed (WS), were provided by Shanghai Meteorological Bureau at Century Park station (located around 2 km away from PEMC). All the above online measurement results were averaged to a 1 h resolution.

## 2.3 Satellite data

Satellite-based visible band sensors are capable of detecting atmospheric aerosols and its spatial distribution in a reliable way (van Donkelaar et al., 2010; Martin, 2008). The aerosol optical depth (AOD) is a widely accepted aerosol index retrieved from satellite sensors. Here we use the AOD measured by the moderate resolution imaging spectrometer (MODIS) on-board the Terra satellite to analyze the long-term evolution of aerosol pollution in Shanghai urban areas during the recent decade (2003-2013). Previously, MODIS-derived AOD data had been well validated with the sunphotometer CE318 measurements at 7 sites over the YRD region (including Pudong site in Shanghai; He et al., 2010). Specifically, the MODIS Level 2 aerosol product (MYD04_3K) was used in this study due to its finer resolution of 3 km and thus greater suitability for providing urban (represented by a 2 km*2 km square centered at $31.22^0$ N, $121.46^0$ E) data. The AOD values were determined with the dark target algorithm, and only high-quality (quality flag=3) data were retained. Detailed information regarding MODIS-derived AOD data processing has been given elsewhere (Cao et al., 2016).

## 2.4 Source identification

Identifying the presence and characteristics of different sources of air pollution is important if air pollution is to be effectively controlled. Two tools, i.e., the bivariate polar plots (BPP) and the potential source contribution function (PSCF), were used in this study to identify the sources and dispersion characteristics of carbonaceous aerosols in Shanghai. The wind speed (WS) and wind direction (WD) in BPP are measured at near ground level, while the WS and WD in PSCF are calculated at a much higher height (typically 500 m a.g.l). Therefore, BPP is more suitable for tracing the origins of air masses at city scale.

Bivariate polar plots demonstrate how the concentration of a targeted species varies synergistically with wind direction and wind speed in polar coordinates, which have proved to be an effective diagnostic technique for discriminating different emission sources (Carslaw and Ropkins, 2012). To construct BPP, wind speed, wind direction and concentration data are firstly partitioned into wind speed-direction 'bins' and the mean concentration calculated for each bin. We set 10 °and 30 for the intervals of wind direction and wind speed, respectively. Given t the inherent wind direction variability in the atmosphere, concentration data with longer time series (e.g., several months in this study) are needed to construct a BPP in order to obtain useful correlations of air concentrations with either wind direction or speed. The two components of wind, $u$ and $v$ are calculated through

$$u = \bar{u} \cdot \sin\left(\frac{2\pi}{\theta}\right), v = \bar{u} \cdot \cos\left(\frac{2\pi}{\theta}\right) \qquad (1)$$

where $\bar{u}$ is is the mean hourly wind speed and $\theta$ is the mean wind direction in degrees with 90 °as being from the east.

Although a $u$, $v$, concentration ($C$) surface can be provided, a better approach is to simulate the surface to describe the concentration as a function of $u$ and $v$ so that we can extract the real source features rather than noise. A mathematical framework for fitting a surface is to use a Generalized Additive Model (GAM) (Wood, 2011), expressed as shown in Equation (2):

$$\sqrt{C_i} = \beta_0 + s(u_i, v_i) + \varepsilon_i \qquad (2)$$

where $C_i$ is the $i$th pollutant concentration, $\beta_0$ is the overall mean of the response, $s(u_i, v_i)$ is the isotropic smooth function of the $i$th value of covariates $u$ and $v$, and $\varepsilon_i$ is the $i$th residual. A penalized regression spline was used to model the surface as described by Wood (2011). It should be noted that $C_i$ is square-root transformed as the transformation generally produces better model diagnostics e.g. normally distributed residuals. Moreover, the smooth function used is isotropic because $u$ and $v$ are on the same scales. The isotropic smooth avoids the potential difficulty of smoothing two variables on different scales e.g. wind speed and direction, which introduces further complexities. The methods described above have been made in the R 'openair' package and are freely available at www.openair-project.org (Carslaw and Ropkins, 2012).

The potential source contribution function is a tool essentially based on air mass back trajectory analysis, which has been widely used to estimate the transporting areas of air pollutants over long distances. In this study, the 24 h back trajectories arriving at Pudong Environmental Monitoring Center at a height of 500 m were calculated at 1 h time intervals for each of the four seasons using the NOAA Hybrid Single Particle Lagrangian Integrated Trajectory (HYSPLIT) model with Global Data Assimilation System (GDAS) one-degree archive meteorological data (Stein et al., 2015). For each trajectory, it includes a range of latitude-longitude coordinates every 1 h backward in a whole day. If the end point of a trajectory falls into a grid cell $(i, j)$, the trajectory is assumed to collect material emitted in the cell (Polissar et al., 1999). The number of end points falling into a single grid cell is $n_{ij}$. Some of these trajectory end points are associated with the data with the concentration of aerosol species higher than a threshold value. The number of these points is $m_{ij}$. The PSCF then calculates the ratio of $m_{ij}$ to the total number of points ($n_{ij}$) in the $ij^{th}$ grid cell. Higher PSCF values indicate higher potential source contributions to the receptor site. Here the domain for the PSCF was set within the range of (26-42$^0$ N, 112.5-125.5$^0$ E) in $0.1^0 \times 0.1^0$ grid cells (Chang et al., 2016). The 75$^{th}$ percentile for OC during the four seasons was used as the threshold value $m_{ij}$. To reduce the uncertainties of $m_{ij}/n_{ij}$ for those grid cells with a limited number of points, a weighting function recommended by Polissar et al. (2001) was applied to the PSCF in each season:

$$w_{ij} = \begin{cases} 1.00, 80 < n_{ij} \\ 0.70, 200 < n_{ij} \leq 80 \\ 0.42, 10 < n_{ij} \leq 20 \\ 0.05, n_{ij} \leq 10 \end{cases} \qquad (3)$$

**3 Results and discussions**

**3.1 Data availability and validation**

Between 10 June 2010 and 31 December 2014, the Sunset carbon analyzer was successfully operated during 75% of the time. The original hourly OC and EC concentrations were judged according to the data before and after the measurement event. Outliers of OC and EC were excluded when it was ten times higher than the nearest two-time points. At least two thirds of the data, i.e., 16 hours in a day, 20 days in a month, and 2 months in a season, must be available so that we can calculate the daily, monthly, and seasonal variations, respectively. In this regard, 28490 hourly data in 39250 hours, or 72.6% data

availability was reached in the current study. Almost no data were collected in March, April, May, and December 2012, March 2013, and April 2014 due to instrument maintenance and malfunction. Therefore, the monthly variations of OC and EC for these six months and seasonal statistics in 2012 Spring (March, April and May) were not considered.

Four seasons in Shanghai were defined as follows: 1 March-31 May as spring, 1 June-31 August as summer, 1 September-30 November as fall, and 1 December-30 December and 1 January-28 February as winter. The mass concentrations of both EC and BC contribute a similar fraction of the carbonaceous aerosol and are supposed to be comparable. The Aethalometer is one of the most frequently utilized techniques to measure real-time BC mass concentrations, especially for long-term background

measurements. In the current study, seasonally different EC concentrations measured by the Sunset OCEC analyzer were validated by comparing with BC data obtained from co-located Aethalometer measurements. The seasonal relationships between Aethalometer BC and Sunset EC are reported in Fig. 2. Slopes ranging from 0.59 to 0.77 are observed with reasonably high correlations ($r > 0.71$). The observed slope indicates that the Aethalometer BC levels were higher than the Sunset EC concentrations.

This might be attributed to the fact that the Aethalometer BC is measured and defined differently than the Sunset EC, which is consistent with earlier studies (e.g., Venkatachari et al., 2006). A higher degree of scatter is observed during the Spring compared to the other seasons, suggesting different thermal, optical and chemical behavior during springtime in our study period.

**3.2 Temporal evolution of OC and EC mass concentrations**

**3.2.1 Data overview**

Summary statistics for the OC and EC concentrations ($\mu g\ m^{-3}$) during 10 July 2010-31 December 2014 are presented in Table 1. Taking all data (n = 28490) as a whole, hourly concentration levels of OC and EC range from 0.20 to 62.05 $\mu g\ m^{-3}$ (average: 7.19 $\pm$4.98 $\mu g\ m^{-3}$) and from 0.06 to 20.49 $\mu g\ m^{-3}$ (average: 2.37 $\pm$1.87 $\mu g\ m^{-3}$), respectively, contributing on average ~20% (OC) and ~5.0% (EC) to the $PM_{2.5}$ mass

between 2013 and 2014. The ratio of OC-to-EC throughout our study period is 3.70 $\pm$ 1.91, which is much high than previous study performed in the Yangtze River Delta region (e.g., 2.36 in Nanjing City (Chen et al., 2017a)). A much higher OC/EC ratio in our study indicates that secondary organic aerosol

is more important in Shanghai, which will be extensively discussed in our next work regarding elucidating its sources and formation mechanism.

Figure 3 shows the mass fractions of hourly carbonaceous aerosols classified by $PM_{2.5}$ levels during 2013 and 2014. Based on a data sample size of n=11790, an approximate lognormal distribution is derived for frequency of $PM_{2.5}$ concentrations. Among them, 56% and 23% of observed $PM_{2.5}$ mass concentrations exceeded China's first-grade and the second-grade National Ambient Air Quality Standard of 35 μg m$^{-3}$ and 75 μg m$^{-3}$, respectively, reflecting heavy aerosol pollution in Shanghai. Carbonaceous aerosols contribute a small fraction (22%) of $PM_{2.5}$ when $PM_{2.5}$ mass concentration greater than 140 μg m$^{-3}$. However, carbonaceous aerosols dominate the chemical components of $PM_{2.5}$ ($\approx$ 50%) when $PM_{2.5}$ concentration lower than 30 μg m$^{-3}$. The result also indicates that the rapid increase in other compounds like inorganic aerosol contributes significantly to heavy haze events in Shanghai, which has also been found in many other cities of China (Huang et al., 2014).

### 3.2.2 Interannual variations

Annually, the OC and EC concentrations (mean $\pm$ 1 $\sigma$) in 2013 (8.9±6.2 and 2.6±2.1 μg m$^{-3}$, n=5547) and 2014 (7.8±4.6 and 2.1±1.6 μg m$^{-3}$, n=6914) were higher than that of 2011 (6.3±4.2 and 2.4±1.8 μg m$^{-3}$, n=8039) and 2012 (5.7±3.8 and 2.0±1.6 μg m$^{-3}$, n=4459). In the first quarter of 2013, China (including Shanghai) experienced extremely severe and persistent haze pollution, with record-high $PM_{2.5}$ mass concentrations (> 500 μg m$^{-3}$) and lasting for days or even a month (Zhang et al., 2015b). It seems that previous pollution control policies (Huang et al., 2013; Chen et al., 2014; Normile, 2008) are ineffective. As indicated in Fig. 4, 2013 had the highest frequency with OC (40% of the time) and EC (43% of the time) loadings higher than 8 μg m$^{-3}$ and 2.5 μg m$^{-3}$, respectively, even though only 41.3% of data was available during January 2013, the most severely polluted month that has been well discussed previously (e.g., Huang et al., 2014; Zhang et al., 2015a). Therefore, a higher carbonaceous aerosol loading in 2013 may be an exception due to the unusually strong influence of haze pollution during wintertime of 2013, which has also been validated previously (Huang et al., 2014).

There is a need to put our study into a longer time frame so that we can obtain a panoramic view of the inter-annual evolution of carbonaceous aerosols levels in Shanghai. Starting from 1999, filter-based measurements of OC and EC were sporadically performed by different groups in Shanghai (SI Table SI),

and all these data together with results from our on-line measurements (line-collected circles) were compiled and depicted in Fig. 5a. It is clear that both OC and EC concentrations in Shanghai show a general downtrend during recent decades, suggesting the success of introducing air-cleaning measures such as greater adoption of renewable energy and raising standards for vehicle emissions (Ji et al., 2012; Ke et al., 2017). However, OC and EC monitoring data between 2000 and 2005 were missing, which make it is impossible to pinpoint the year of transition into a decreasing pattern. More importantly, significant differences (e.g., a factor of two) have been reported for levels of OC and EC when comparing various analytical techniques (Schauer, et al., 2003; Karanasiou, et al., 2015), posing an insurmountable task for us to quantitatively analyze the variations of carbonaceous aerosols and their responses to pollution control measures in Shanghai. As an alternative, we retrieved the AOD data from MODIS satellite in urban and rural Shanghai (Fig. 5b) in order to reproduce the fluctuation of aerosol loading between 2003 and 2013. Figure 4b reinforces a growing consensus that the year of 2006 marked a milestone for Shanghai acting as a pioneer in terms of replacing coal with nature gas. Over 90% of $SO_2$ emissions in China were derived from coal combustion (Lu et al., 2011), and ambient $SO_2$ concentrations were directly related to $SO_2$ emissions. Therefore, the long-term evolution of $SO_2$ concentration in SI Fig. S2 reflects the process of replacing coal with nature gas in Shanghai. Indeed, energy consumption structure in China, for long stretches, is overwhelmingly dominated by coal despite the fact that PM emissions from natural gas burning can be negligible to a large extent when compared with coal combustion (Hayhoe et al., 2002). Benefitting from China's west-east gas pipeline project (Fig. 1), PM emissions in Shanghai have been cut immensely as there are 7.5 million gas users, including more than 5000 industrial users, and 80% of city taxies are fueled by gas (Hao et al., 2016; Huo et al., 2013). In conclusion, the results from field measurements and satellite observations in Fig. 5 jointly give a "ground truth" that carbonaceous aerosols loading in Shanghai decreased in general from 2006; this was largely driven by the implementation of the clean energy initiative (Lei et al., 2011; Zhao et al., 2013).

**3.2.3 Monthly and seasonal variations**

The hourly, monthly, seasonal, and annual variations of OC and EC concentrations are illustrated in Fig. 6. The monthly average mass concentrations of carbonaceous aerosols show relatively large variations in this study (Fig. 6), with the average value ranging from 4.0 (September 2010) to 15.5 (December 2013) µg m$^{-3}$ for OC, and from 1.4 (September 2014) to 4.7 (December 2013) µg m$^{-3}$ for EC (Table 1). The

months of September and December presented the lowest (OC: 5.0 μg m$^{-3}$; EC: 1.7 μg m$^{-3}$) and the highest (OC: 9.7 μg m$^{-3}$; EC: 3.5 μg m$^{-3}$) average carbonaceous aerosols throughout our study period. This is similar to most cities across China due to a combination of emissions and seasonal variance in meteorology (Cao et al., 2007). Previous work (Chang et al., 2016) show that the month of September in Shanghai has the highest planetary boundary layer (PBL), which favours the vertical dispersion of ground-emitted pollutants. Besides, higher temperature in September could lead to a shift in the gas-particle equilibrium with more semi-volatile organic compounds (SVOCs) remaining in the gas phase (Yang et al., 2011). In addition, carbonaceous aerosols can also be effectively removed by wet deposition attributed to large amount of precipitation in September (Wang et al., 2006). In contrast, carbonaceous aerosol concentrations elevated in the month of December resulting from relatively stable atmospheric conditions, existence of temperature inversion, and variations in emissions (Feng et al., 2006; Zielinska et al., 2004).

Seasonally, the average concentrations of OC ranged from 5.2 (summer of 2012) to 10.3 (winter of 2014) μg m$^{-3}$, and EC ranged from 1.6 (summer of 2012) to 4.0 (winter of 2010) μg m$^{-3}$. Except for a slightly higher concentration of EC in the fall of 2012 due to the boost of intensive pollution episodes (indicating by a significantly higher value in P95 or 95[th] percentile; Table 1), higher concentrations of EC were observed in winter for other years, which could be caused by the stagnation of the atmosphere and the stronger influence of regional transport during wintertime (Chen et al., 2017a). However, it is worth noting that the concentration levels of EC in spring, summer and fall were overall similar (Table 1), reflecting a generally local-dominated EC emissions in Shanghai (Cao et al., 2007). As shown in Table 1, there is no uniform pattern for the seasonal variations of OC concentrations, which can be explained by their complexity in terms of the formation processes and sources (see detailed discussion in section 3.2.4 and 3.3). Indeed, in contrast to EC, OC is the mixed product of primary emissions and secondary formation, which could vary significantly in different seasons. In summer, for example, strong solar radiation in summer tends to facilitate photochemical reactions and thus enhance the formation of VOCs to organic aerosols (Tuet et al., 2017; Malecha and Nizkorodov, 2016). Burning of crop residues is also an important source of OC in China (Cheng et al., 2014; Hallquist et al., 2009; Zhang and Cao, 2015), while the burning activities are highly seasonal-dependent and mostly concentrated in fall (Chen et al., 2017b). Cold air masses prevail during winter and early spring, transporting high levels of pollutants

(including OC and its precursors emitted from coal-based heating system) from Northern China to the Yangtze River Delta region (Chen et al., 2017a). Scatter plots of OC and EC and their correlation coefficients are shown by season in Fig. 7. Significant correlations ($R^2 \approx 0.70$, $p < 0.001$) between OC and EC were found for winter and fall, indicating that OC and EC share certain similar sources in Shanghai during the two seasons. The highest degree of correlation ($R^2 = 0.71$) was observed during wintertime when the seasonal OC/EC ratio was the lowest (Table 1), suggesting that primary emissions are an important source contributing to both OC and EC in Shanghai during the cold season (see detailed discussion in section 3.3).

### 3.2.4 Diurnal variations and weekend-weekday comparisons

Diurnal variations of EC concentrations during weekdays and weekends in different seasons of each year are shown in Fig. 8. CO concentrations at Pudong site were highly correlated with EC concentrations (Fig. 9). Diurnal variations of OC concentrations during weekdays and weekends in different seasons of each year are shown in Fig. 10, respectively. We first focus on the difference between weekday and weekend patterns. Previously, several short-term studies (e.g., Feng et al., 2009; Stone et al., 2008; Yu et al., 2009; Kim and Hopke, 2008) have observed pronounced differences in the OC and EC patterns between weekdays and weekends, which were thought to be the cycling effects of anthropogenic activities such as vehicular emissions. In Fig. 8, weekday concentrations of EC were 27%, 22%, and 21% higher than the weekend concentrations in the winter of 2011 (the second row from the top and hereafter, the fourth column from the left and hereafter), fall of 2012 (the third row, the second column), and summer of 2013 winter (the fourth row, the fourth column), respectively. We also found that the weekday concentrations of OC in the fall of 2010 (the first row, the third column), fall of 2012 (the third row, the third column) and summer of 2013 (the fourth row, the second column) were 17.8%, 29.7%, and 32.8% higher than that of weekend, respectively (Fig. 10). Nevertheless, our long-term measurement shows that there is no uniform pattern of weekend effect on both OC and EC over the duration of the campaign, suggesting that local sources of carbonaceous aerosols (e.g., on-road traffic) in Shanghai urban center do not vary significantly between weekdays and weekends. To validate our conclusion, we collected traffic flow data (2012-2014) in urban road network from Shanghai Traffic Administration Bureau. SI Fig. S3 shows that average traffic flows in weekends were only around 8% lower than that in weekdays.

Distinctive diurnal patterns are observed for EC, generally characterized by two marked peaks occurring at morning (peaking at around 08:00 local time) and during the evening (starting at around 16:00 local time), consistent with the variation of traffic flow in Shanghai (Chang et al., 2012, 2014, 2016). Moreover, as we stated above, CO concentrations were highly correlated with EC concentrations (Fig. 9), indicating that CO and EC emissions in Shanghai share similar sources (Zhao et al., 2013). Scatter plots of EC vs. CO in Fig. 9 confirms that on-road traffic is a significant source contributing to EC emissions in Shanghai (Chen et al., 2017; Yang et al., 2005).

Relatively flat diurnal cycles were observed for OC during most seasons (Fig. 10). A multi-day build-up of OC was frequently observed during all months (Fig. 4), supporting the notion of regional influences on OC in Shanghai. There was no obvious decrease of OC in the daytime, which might be explained by secondary organic formation. It should be noted that the daytime photochemical production of OC from gas-phase oxidation of VOC might be masked by an elevated planetary boundary layer (PBL). Considering the dilution effect of the PBL height, Sun et al. (2012) found that OC increased gradually from morning to late afternoon, demonstrating the importance of daytime photochemical production of SOA. In this study, OC in fall and winter showed a clear daytime increase until late afternoon, illustrating possible role played by gas-phase photochemical processing in driving the OC diurnal cycle. However, it should be noticed that it is still difficult to pinpoint a specific OC emission source based on our current data. Since ambient OC concentration depends on multiple factors, further study will reveal more information about the different formation mechanisms involved.

Examining the relationship between OC vs. trace gas species ($NO_2$ and $O_3$) could provide more information on the formation and transformation of ambient OC. Since $NO_2$ is a major pollutant emitted from combustion processes, its correlation with OC confirms that combustion-generated carbon emissions (including on-road traffic) are important source of OC emissions in Shanghai. OC vs. $O_3$ does not show obvious correlation (Fig. 11a). Scatter plots of OC vs. $NO_2$ of observations at Pudong site show reasonable correlations (Fig. 11b). This finding concurs with the result observed in Mexico City (Yu et al., 2009). First, although $O_3$ favour the formation of secondary organic aerosols (SOA), OH radical (OH$^\cdot$)-initiated oxidation may dominate SOA formation, including aqueous-phase oxidation and $NO_3$-radical-initiated nocturnal chemistry. Second, ambient $O_3$ concentrations often decreased sharply under

high PM loading (> 100 μg m$^{-3}$), which could inhibit SOA formation through O$_3$ oxidation (Huang et al., 2014).

**3.3 Meteorological effects**

Figure 12 shows the RH- and *T*-dependent distributions of OC and EC mass concentrations throughout the study period. OC show the highest mass loading (> 8 μg m$^{-3}$) at *T* > 30 °C, and has no clear dependence on RH. This is coincident with the RH- and *T*-dependent distribution pattern of OC/EC (Fig. 12c), which will be used to further investigate the RH/*T* impacts on the formation of secondary organic aerosols in future work. Occasionally, high OC mass concentrations can also be found at median and low *T* (Fig. 12a), indicating complex sources of OC in Shanghai. For example, there are also several grids with high OC concentrations in the bottom right of Fig. 12a but low OC/EC ratio (< 2; Fig. 12c), suggesting the important contribution of primary sources to ambient OC under very low *T* (< 0 °C) and high RH (> 80%) in Shanghai. At -1 °C, for instance, the diesel vehicles were estimated to emit 7.6 times more OC than at 21 °C (Zielinska et al., 2004). In contrast to OC, the distribution of EC shows a more obvious concentration gradient as a function of both *T* and RH (Fig. 12b), and this is particularly true for WS (Fig. 12d). In Fig. 12b, EC shows the highest mass loading, generally higher than 3.5 μg m$^{-3}$ at *T* < 0 °C and RH > 80%. This can be explained by the lowest WS (often lower than 1 m s$^{-1}$; Fig. 12d) occurring under low *T* and high RH that tended to accumulate more EC within the city's shallow surface layer (Fig. 12d). In fact, EC and WS demonstrated a generally contrasting RH- and *T*-dependent distribution pattern throughout our study period, which strongly indicates that EC variations in Shanghai were WS-dependent.

In Fig. 13a, EC concentrations show more evident WS gradients than OC, with higher concentrations in association with lower wind speeds. This is in line with the conclusion made above that EC concentrations in Shanghai were more sensitive to WS, highlighting the dominating role of local contribution to EC emissions in Shanghai. However, OC and EC in spring did not vary synchronously with WS at low WS (< 1 m s$^{-1}$) in the spring, suggesting that regional transport can also be an important source contributing to ambient carbonaceous aerosols in Shanghai. For OC concentrations, their variations in other seasons were also not strictly followed WS gradients, confirming that the share of regional transport contribution to ambient OC was comparable with local sources in Shanghai.

In Fig. 13b, both OC and EC show clear wind sector gradients, with higher concentrations in association with winds from the southwest (SW) and west (W), and lower concentrations with east (E) and northeast (NE) wind. The average mass concentrations of OC and EC from the SW were 9.8 μg m$^{-3}$ and 3.4 μg m$^{-3}$, respectively, which were ~1.7 times than that from the NE (5.9 μg m$^{-3}$ and 1.9 μg m$^{-3}$ for OC and EC, respectively). This can be explained by the high WS (often larger than 6 m s$^{-1}$; Fig. 14) from the E and NE associated with clean air masses from remote oceans (SI Fig. S1) while low WS (often lower than 6 m s$^{-1}$; Fig. 14) from the W and SW is associated with polluted air masses from inner Shanghai or inland China (SI Fig. S1). Overall, OC and EC increased as wind sectors changed along the NE-E-SE-S-SW gradient, and then decreased along the SW-W-NW-N gradient (Fig. 13). Such wind sector dependence of carbonaceous aerosols is generally consistent with the spatial distribution of PM$_{2.5}$ in Shanghai and its neighbouring regions (Ma et al., 2014). However, this study identified that winds from the SW were the most important pathway of contributing ambient carbonaceous aerosols from inland China to Shanghai. This result is different from most previous studies which concluded that the NW or N in Northern China were the dominant source region of coal-related aerosol species (such as sulfate) and gases (such as SO$_2$).

Seasonal bivariate polar plots (BPP) of EC concentrations for 2010-2014 are shown in Fig. 14, and the distribution pattern generally agrees with OC (SI Fig. S4). There are several points that should be noted. First, higher EC mass loadings (> 3 μg m$^{-3}$) but lower OC/EC ratios (< 3; SI Fig. S5) near the field site (indicating by WS < 2 m s$^{-1}$) are seen as a characteristic feature for every season throughout the sampling period, suggesting that local and primary emissions (e.g., on-road traffic) are a stable and important source contributing to ambient EC concentrations in Shanghai. Second, Shanghai's main urban districts are southwest of the field site (SI Fig. S1). Fig. 13b and Fig. 14 jointly identify that winds from the SW consistently had the highest EC concentrations. Given the relatively low WS (< 6 m s$^{-1}$ in most cases; Fig. 14) in the SW (where the man urban districts located), the urban areas of Shanghai were highly likely the major source areas for EC. Thirdly, EC also displayed relatively high concentrations in the NW during winter and spring, and the lowest in the E and SE during summer and fall (Fig. 14). In fact, Shanghai has a subtropical monsoon climate and is bounded to the east by the East China Sea (SI Fig. S1). Winters and springs in Shanghai are chilly, with northwesterly winds from Siberia transporting high levels of air pollutants in Northern China to the YRD region resulting in poor air quality in Shanghai. Conversely, the SE winds originating from the East China sea prevail in summers and fall. Moreover, the city is

susceptible to typhoons in summer and the beginning of fall, thus carbonaceous aerosols can be effectively removed by wet deposition attributed to large amount of rainfall (Huang et al., 2008).

As shown in Fig. 15, the OC concentrations in Shanghai also have potential source areas in common. There is a pollution transport belt along the middle and lower reaches of the Yangtze River among the four seasons, which can be attributed to two main reasons. First, there are five important rice-growing areas (of a total of nine in China), i.e., the Jianghan Plain in Hubei Province, Dongting Lake Plain in Hunan Province, Poyang Lake Plain in Jiangxi Province, Yangtze-Huaihe area in Anhui and Jiangsu Province, and Taihu Lake Plain in Jiangsu Province are under the pollution transport belt (SI Fig. S1). Previously, Field measurements, model simulations and satellite observations have identified that inter-province transport of air pollutants emitted from un-prescribed biomass (mainly rice residues) burning was an important source of air pollution throughout the YRD region. For example, based on ambient monitoring data and the WRF/CMAQ (Weather Research and Forecasting (WRF) and Community Multiscale Air Quality (CMAQ)) model simulation during the beginning of summer in 2011, Cheng et al. (2014) showed that biomass burning contributed 37% of $PM_{2.5}$, 70% of OC and 61% of EC in five cities (including Shanghai) of the YRD region. Moreover, it is estimated that the $PM_{2.5}$ exposure level could be reduced by 47% for the YRD region if complete biomass burning was forbidden and a significant health benefit would be expected (Cheng et al., 2014). Nevertheless, biomass burning activities are mostly concentrated in the months of June and October, the periods of crop post-harvest. The second but more important reason could be the influence of anthropogenic activities (excluding prescribled biomass burning; Chen et al., 2017a). Indeed, the pollution belt overlaps the Yangtze River Economic Belt, one of China's most densely populated areas clustering with many cities (including big cities like Wuhan, Changsha, Nanchang, Hefei, Nanjing, and Suzhou) and numerous industrial complexes (including massive petrochemical, iron and steel, and chemical processing industry). In brief, our PSCF analysis highlighted the importance of long-range transport contributing to OC pollution in Shanghai.

**4 Conclusions**

This paper presents the results from a multi-year and near real-time measurement study of carbonaceous aerosols in $PM_{2.5}$ using a Sunset semi-continuous OC/EC analyser, conducted at an urban site in Shanghai

from July 2010 to December 2014. The annual mass concentrations of OC (EC) from 2011 to 2014 were 6.3±4.2 (2.4±1.8), 5.7±3.8 (2.0±1.6), 8.9±6.2 (2.6±2.1), and 7.8±4.6 (2.1±1.6) µg m$^{-3}$, respectively, accounting for 13.2-24.6% (3.9-6.6%) of PM$_{2.5}$ mass. We integrated the results from historical field measurements and satellite observations, concluding that carbonaceous aerosol pollution in Shanghai has gradually reduced since 2006. Our results confirm the success of replacing coal with cleaner energy such as natural gas in Shanghai, which can be adopted in other megacities like Beijing and Guangzhou to curb PM$_{2.5}$ pollution.

Both OC and EC showed concentration gradients as a function of wind direction and wind speed, generally with higher values associated with winds from the southwest, west, and northwest. This was consistent with their higher potential as source areas, as determined by the potential source contribution function analysis. A common high potential source area, located along the middle and lower reaches of the Yangtze River instead of Northern China, was pinpointed during all seasons. The results of this study also highlighted that the reduction of biomass burning and anthropogenic emissions for the YRD region requires regional joint management and control strategies.

**5 Data availability**

Data are available from the corresponding authors on request. The authors prefer not to publish the data at the present stage in order to avoid compromising the future of ongoing publications.

*Acknowledgements*. This study was supported by the National Key Research and Development Program of China (2017YFC0210100), National Science Foundation of Jiangsu Province (BK20170946), University Science Research Project of Jiangsu Province (17KJB170011), National Science Foundation of China (Grant nos. 91644103 and 41603104). Yunhua Chang and Zhong Zou acknowledge the support of the Start-up Foundation for Introducing Talent to NUIST and Shanghai Pudong New Area Sci-tech Development Funds (Grant no. PKJ2016-C01), respectively. We also acknowledge the Qingyue Open Environmental Data Centre (http://data.epmap.org) for the unconditional help in terms of providing criteria pollutants monitoring data.

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

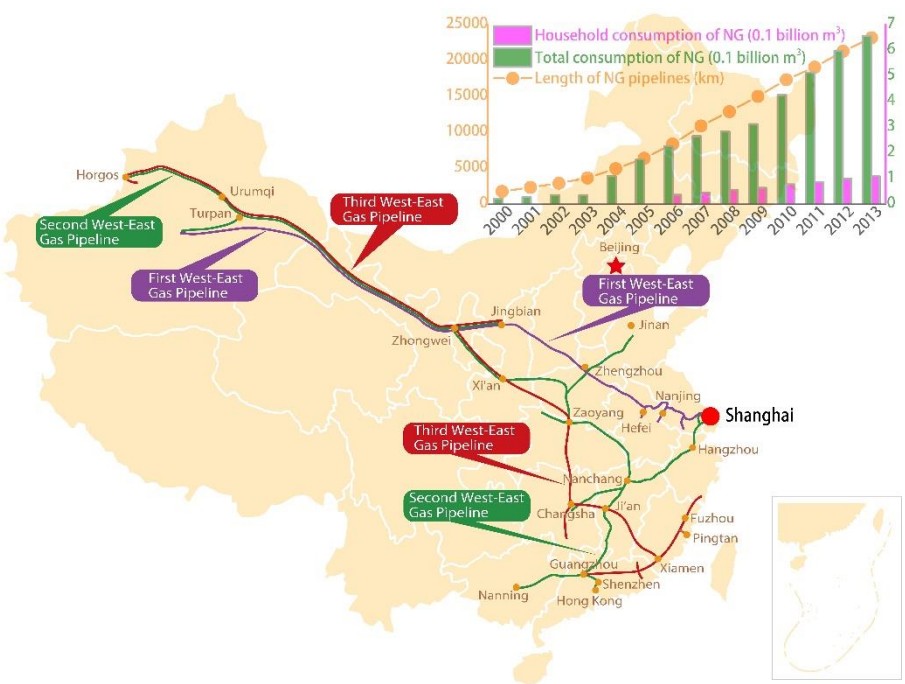

**Figure 1.** Every pipeline leads to Shanghai. The west-east pipeline project (WEPP) is a set of natural gas pipelines which transport clean natural gas from Xinjiang to the energy-hungry Yangtze River Delta region (including Shanghai). Started in 2002, the construction of the WEPP is one of China's largest energy infrastructures. The first, second, and third pipelines were completed in 2005, 2011, and 2015, respectively. The figure at the top right shows the rapid increase of pipeline construction and natural gas consumption in Shanghai between 2000 and 2013 (Wen, 2014).

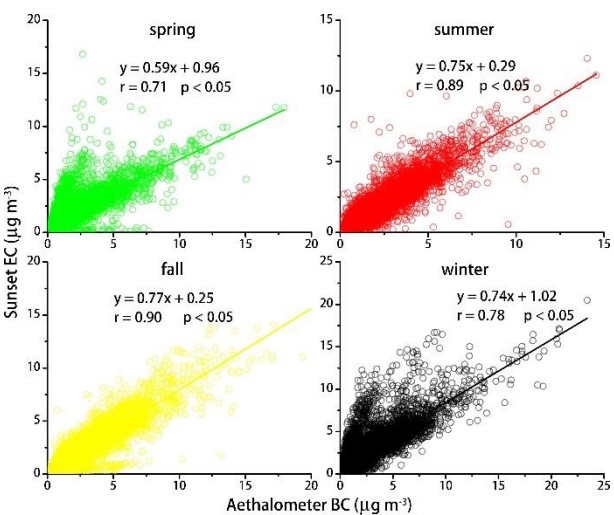

**Figure 2.** Comparison between collocated Aethalometer BC and Sunset EC concentrations for different seasons.

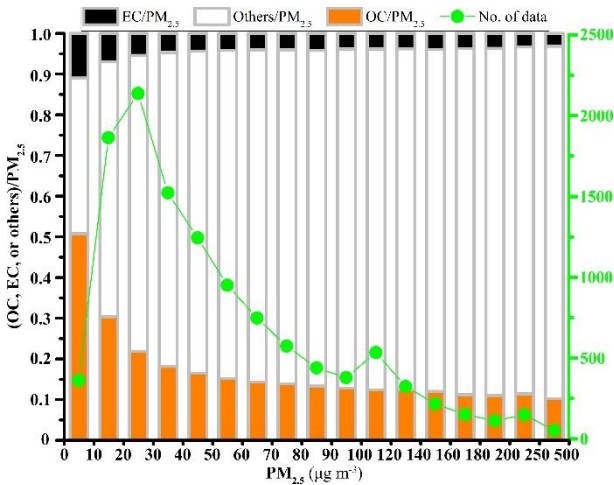

**Figure 3.** The contribution fraction of OC, EC, and other aerosol species to ambient $PM_{2.5}$ at different

$PM_{2.5}$ concentration intervals in Shanghai between January 2013 to December 2014.

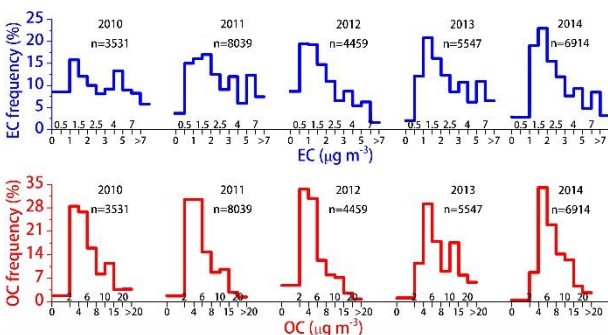

**Figure 4.** Frequency of EC and OC mass loadings in five different years in Shanghai. The frequency was calculated based on the average data points within a mass concentration internal of 0.5 μg m$^{-3}$ and 2 μg m$^{-3}$ for EC and OC, respectively.

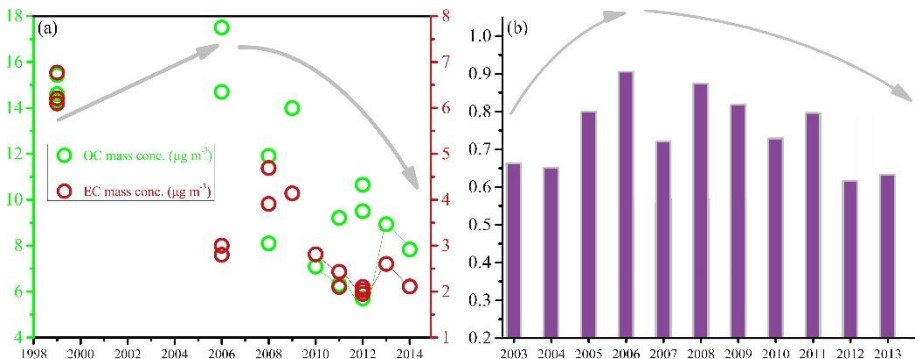

**Figure 5.** Long-term evolution of annual carbonaceous aerosol concentrations **(a)** and MODIS-derived aerosol optical depth **(b)** in Shanghai. Note that the line-collected circles in the left figure represent the results in this study, the sources of the rest circles are listed in SI Table S1.

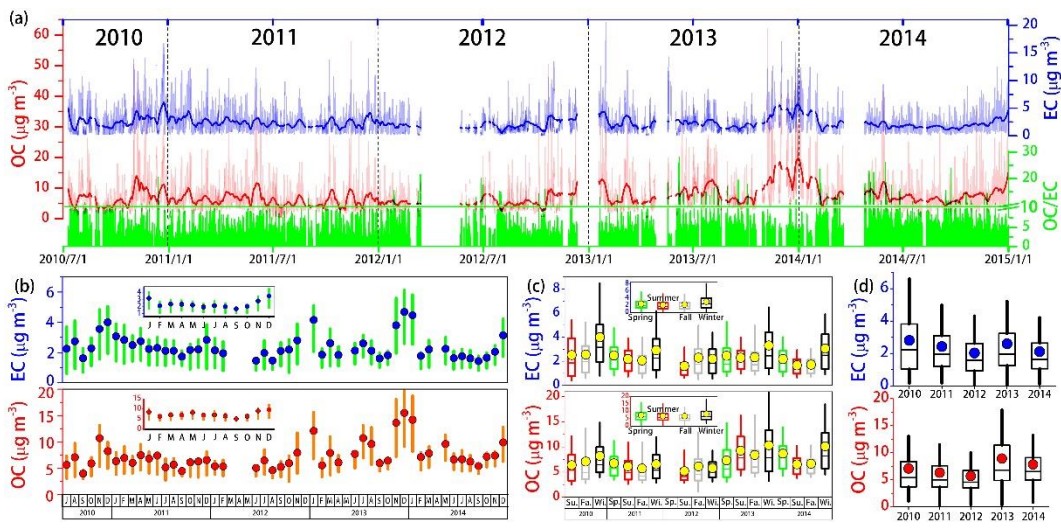

**Figure 6.** Hourly **(a)**, monthly **(b)**, seasonal **(c)**, and interannual **(d)** variations of OC and EC mass concentrations in Shanghai. Note that the variations of OC/EC are also shown in **a**, setting 10 as the breaking point. The relatively small figures in **b** and **c** represent the overall monthly and seasonal mass concentrations of OC and EC throughout our study period, respectively.

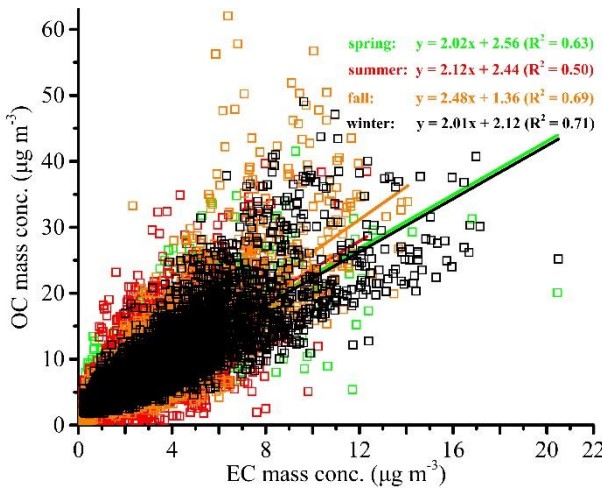

**Figure 7.** Correlation between EC and OC mass concentrations during different seasons in Shanghai.

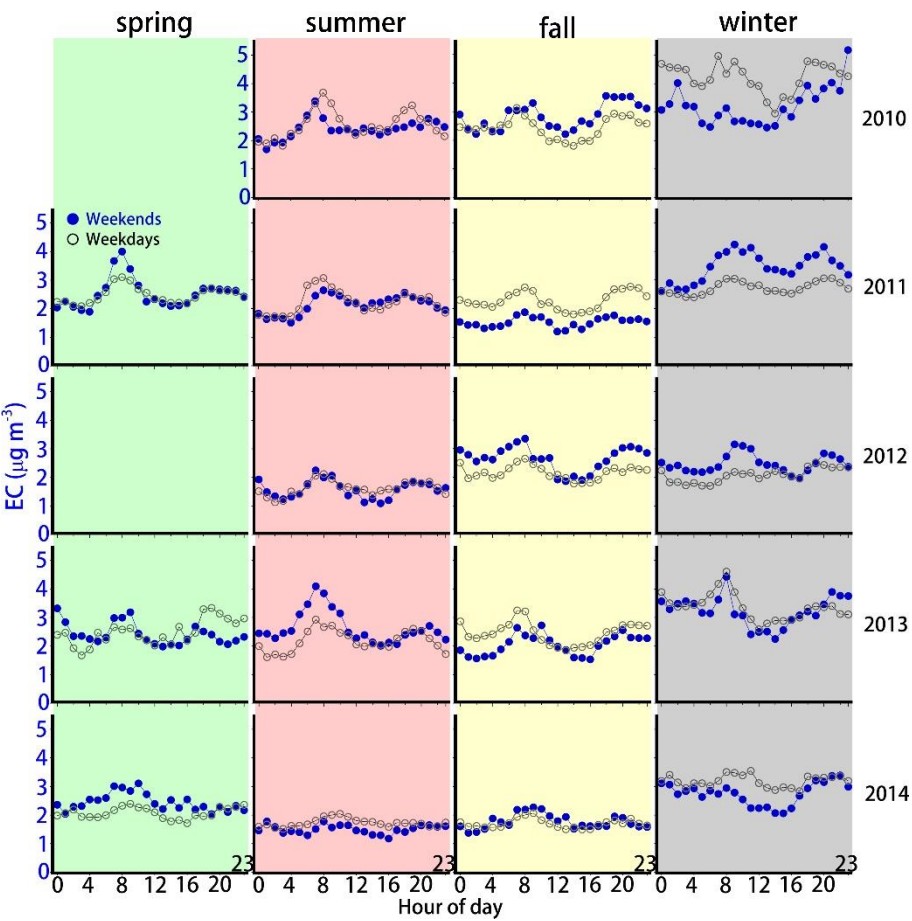

**Figure 8.** Diurnal variations of EC concentrations during weekdays and weekends over different years in Shanghai.

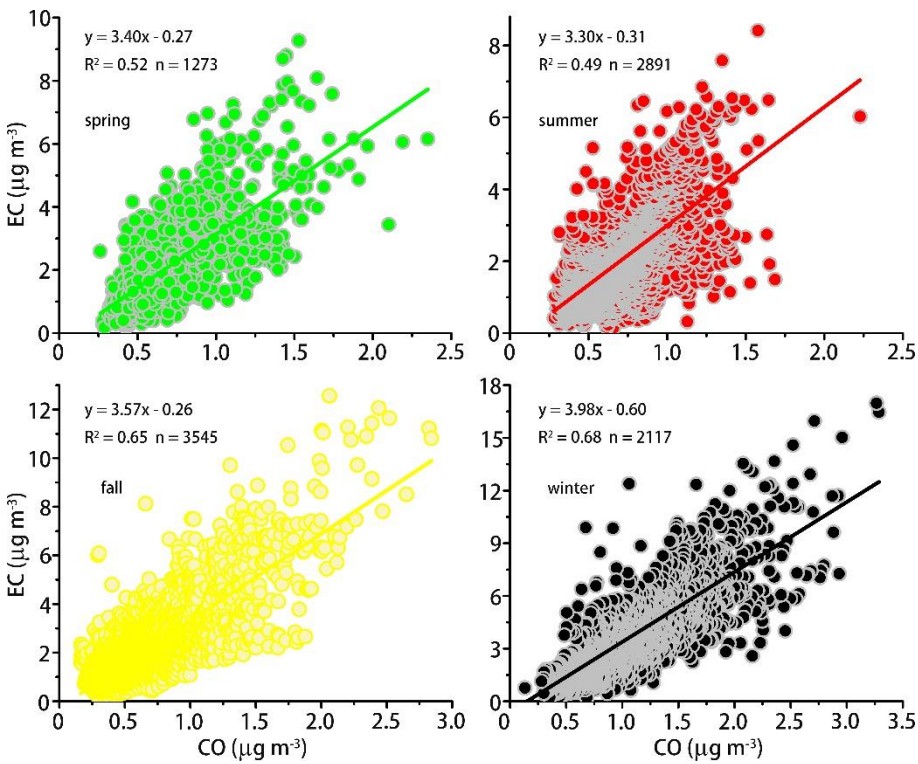

**Figure 9.** Correlation analysis of the mass concentrations of CO and EC during four seasons in Shanghai.

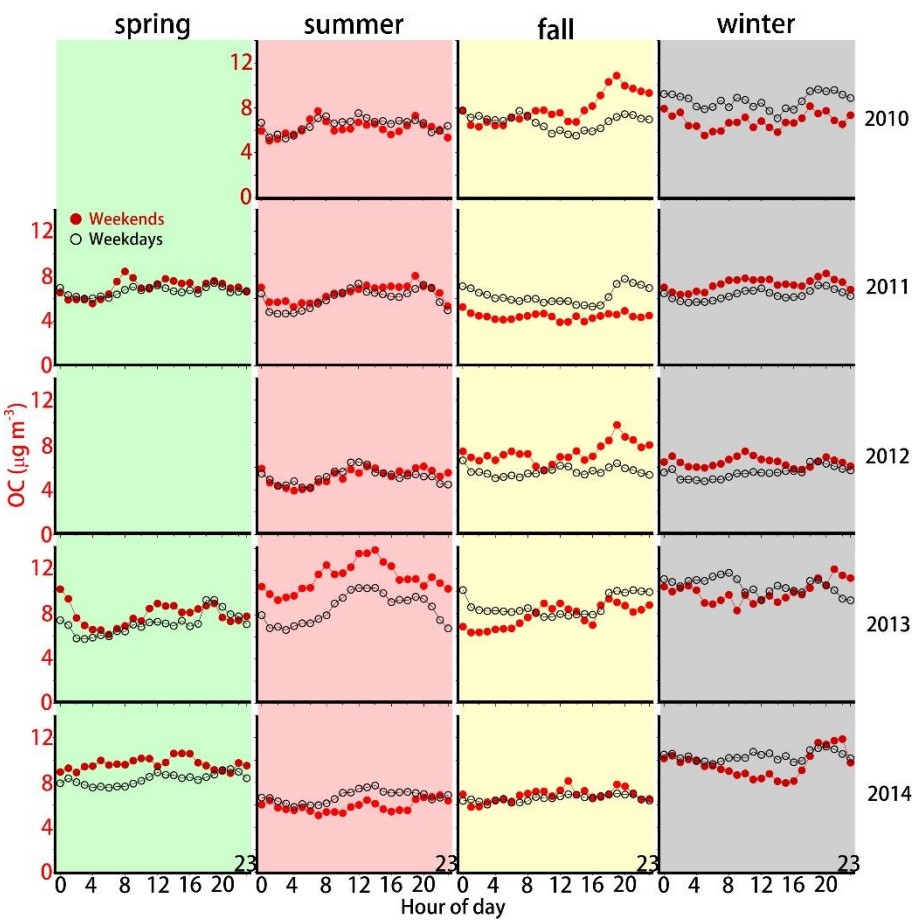

**Figure 10.** Diurnal variations of OC concentrations during weekdays and weekends over different years in Shanghai.

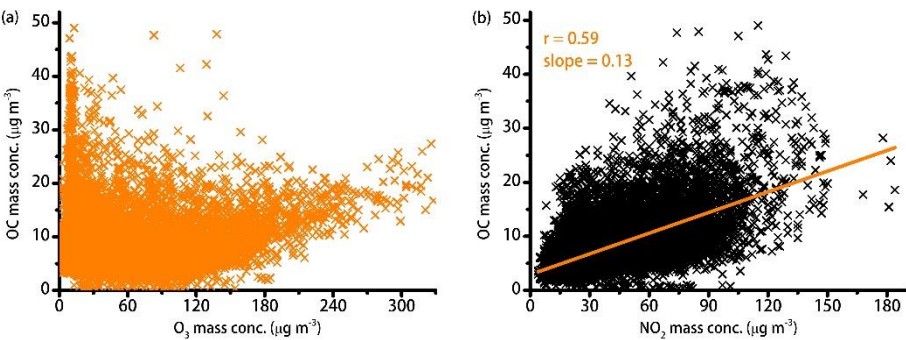

**Figure 11.** Scatter plots of OC vs. O₃ (**a**) and OC vs. NO₂ (**b**) in Shanghai.

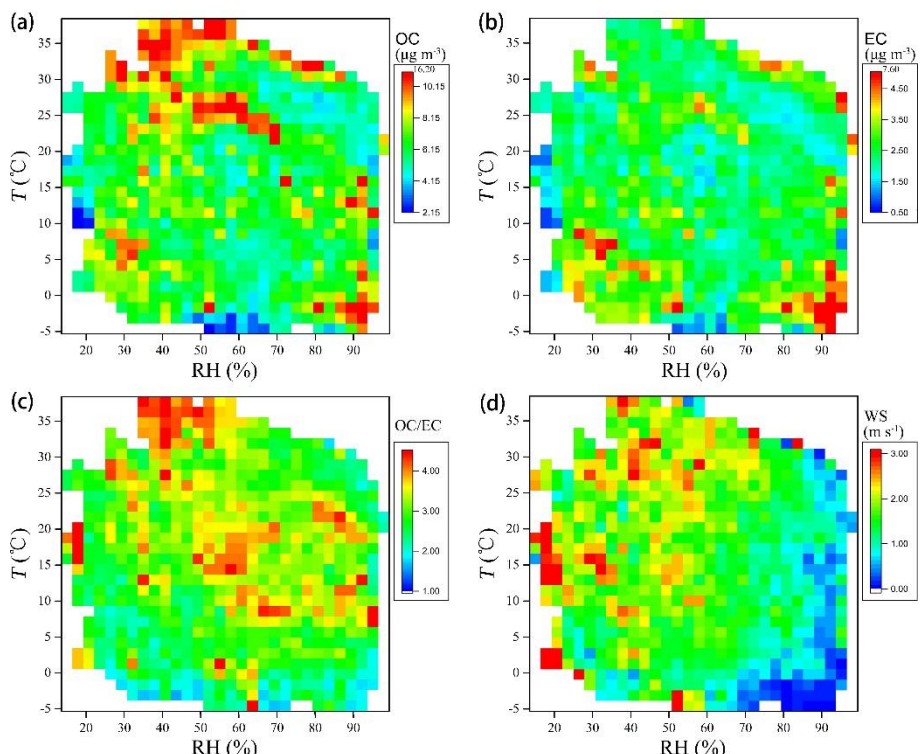

**Figure 12.** RH/*T* dependence of **(a)** OC mass concentrations, **(b)** EC mass concentrations, **(c)** OC/EC ratios, and **(d)** wind speeds (WS) between July 2010 and December 2014 in Shanghai. The data are grouped into 900 (30*30) grids with increments of RH and *T* being 2.86% and 1.46 °C, respectively.

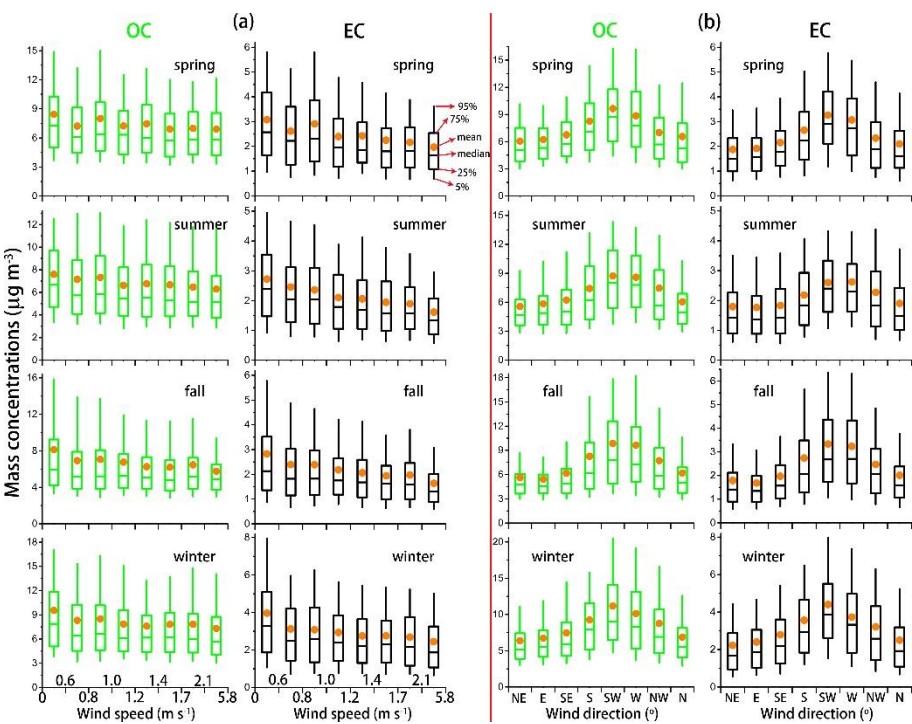

**Figure 13.** Box plots of OC and EC mass concentrations as a function of wind speed increments **(a)** and wind direction sectors **(b)** between July 2010 and December 2014 in Shanghai. The mean, median, 5th, 25th, 75th and 95th percentiles are indicated in the second figure of the top row.

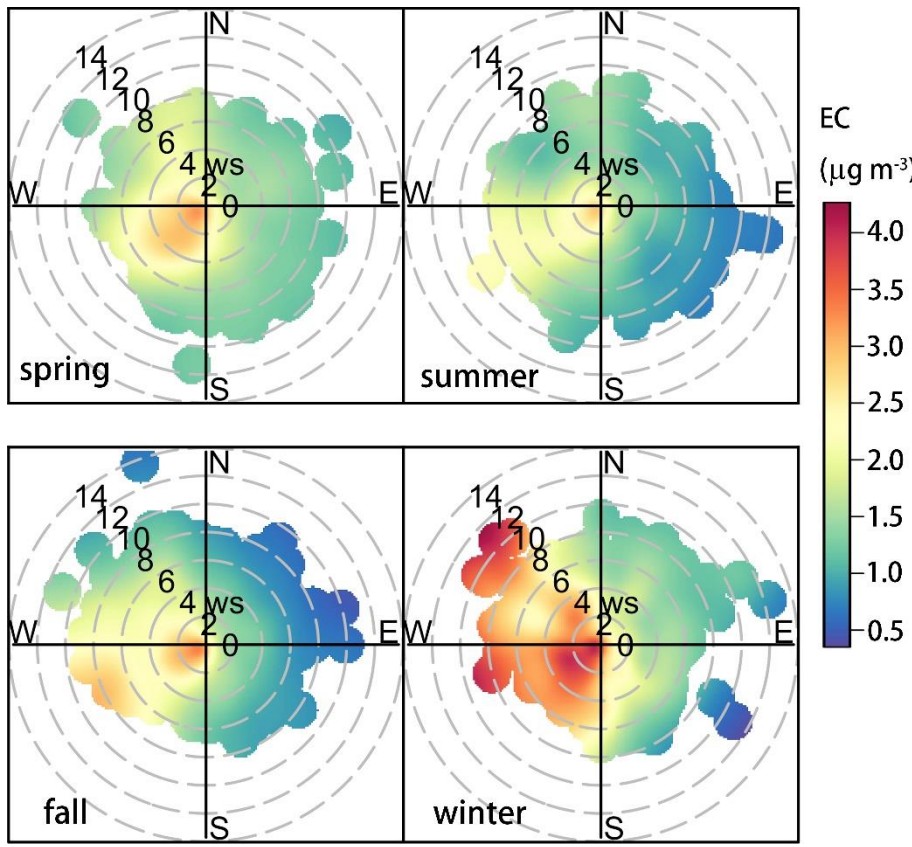

**Figure 14.** Bivariate polar plots of seasonal EC concentrations (μg m$^{-3}$) in Shanghai between July 2010 and December 2014. The center of each plot represents a wind speed of zero, which increases radially outward. The concentration of EC is shown by the colour scale.

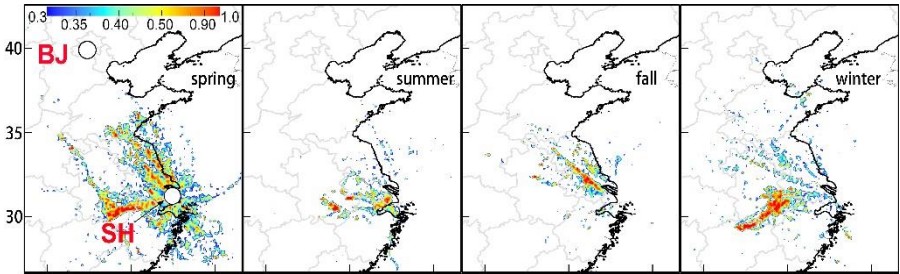

**Figure 15.** Potential Source Contribution Function (PSCF) of OC during four seasons in 2014 in Shanghai. The cities marked in the leftmost panel are Beijing (BJ) and Shanghai (SH). The colour scales indicate the values of PSCF.

**Table 1.** Descriptive statistics of the annual and seasonal variations of EC, OC, and OC/EC during 10 July 2010-31 December 2014 in Shanghai. Note that the statistics result of 2012 spring is not given due to data deficiency.

|  | Year | Season | N | Mean | SD | Min | P5 | Q1 | Median | Q3 | P95 | Max |
|---|---|---|---|---|---|---|---|---|---|---|---|---|
|  |  | annual | 3531 | 2.81 | 2.39 | 0.17 | 0.39 | 1.03 | 2.23 | 3.82 | 7.41 | 16.72 |
|  | 2010 | Summer | 1050 | 2.52 | 2.02 | 0.17 | 0.32 | 0.76 | 1.87 | 3.92 | 6.32 | 10.68 |
|  |  | Fall | 1846 | 2.56 | 2.12 | 0.24 | 0.45 | 1.06 | 1.96 | 3.28 | 6.89 | 14.07 |
|  |  | Winter | 635 | 4.00 | 3.18 | 0.22 | 0.57 | 1.93 | 3.17 | 5.06 | 11.48 | 16.72 |
|  |  | annual | 8039 | 2.43 | 1.80 | 0.19 | 0.55 | 1.21 | 1.94 | 3.10 | 6.12 | 13.94 |
|  |  | Spring | 2128 | 2.49 | 1.61 | 0.23 | 0.64 | 1.35 | 2.12 | 3.16 | 5.80 | 12.11 |
|  | 2011 | Summer | 1903 | 2.19 | 1.40 | 0.22 | 0.60 | 1.18 | 1.86 | 2.91 | 4.74 | 12.32 |
|  |  | Fall | 1940 | 2.07 | 1.62 | 0.19 | 0.50 | 1.02 | 1.69 | 2.47 | 5.37 | 13.68 |
|  |  | Winter | 2068 | 2.92 | 2.29 | 0.22 | 0.50 | 1.25 | 2.28 | 3.86 | 7.84 | 13.94 |
|  |  | annual | 4459 | 2.03 | 1.59 | 0.07 | 0.37 | 0.93 | 1.57 | 2.62 | 5.31 | 13.47 |
|  |  | Spring | 192 |  |  |  |  |  |  |  |  |  |
| EC | 2012 | Summer | 1259 | 1.61 | 1.19 | 0.16 | 0.53 | 0.87 | 1.23 | 1.88 | 4.05 | 9.86 |
|  |  | Fall | 1520 | 2.30 | 1.89 | 0.07 | 0.28 | 1.02 | 1.75 | 3.02 | 6.11 | 13.47 |
|  |  | Winter | 1488 | 2.20 | 1.52 | 0.22 | 0.45 | 1.00 | 1.86 | 3.00 | 5.31 | 8.70 |
|  |  | annual | 5547 | 2.60 | 2.07 | 0.06 | 0.70 | 1.26 | 1.97 | 3.24 | 6.60 | 20.49 |
|  |  | Spring | 1080 | 2.46 | 2.13 | 0.06 | 0.48 | 1.13 | 1.74 | 3.08 | 6.56 | 20.46 |
|  | 2013 | Summer | 1570 | 2.29 | 1.31 | 0.16 | 0.77 | 1.34 | 2.03 | 2.89 | 4.85 | 8.41 |
|  |  | Fall | 1465 | 2.33 | 1.84 | 0.29 | 0.76 | 1.20 | 1.72 | 2.71 | 6.46 | 12.56 |
|  |  | Winter | 1432 | 3.32 | 2.68 | 0.20 | 0.72 | 1.41 | 2.46 | 4.39 | 8.63 | 20.49 |
|  |  | annual | 6914 | 2.11 | 1.55 | 0.12 | 0.60 | 1.06 | 1.64 | 2.64 | 5.24 | 12.41 |
|  |  | Spring | 1284 | 2.19 | 1.43 | 0.18 | 0.57 | 1.14 | 1.83 | 2.86 | 5.10 | 9.27 |
|  | 2014 | Summer | 1829 | 1.67 | 0.98 | 0.15 | 0.52 | 0.96 | 1.43 | 2.17 | 3.58 | 6.17 |
|  |  | Fall | 2144 | 1.72 | 1.09 | 0.12 | 0.58 | 0.99 | 1.43 | 2.14 | 3.98 | 9.01 |
|  |  | Winter | 1657 | 3.06 | 2.14 | 0.21 | 0.77 | 1.35 | 2.50 | 4.25 | 7.12 | 12.41 |

| | | | | | | | | | | | | |
|---|---|---|---|---|---|---|---|---|---|---|---|---|
| | | annual | 3531 | 7.09 | 5.50 | 1.06 | 2.41 | 3.69 | 5.39 | 8.38 | 17.74 | 50.46 |
| | 2010 | Summer | 1050 | 6.39 | 4.00 | 1.06 | 2.07 | 3.31 | 5.36 | 8.40 | 14.02 | 26.32 |
| | | Fall | 1846 | 7.09 | 6.28 | 1.66 | 2.47 | 3.56 | 4.92 | 7.69 | 20.11 | 50.46 |
| | | Winter | 635 | 8.22 | 5.04 | 1.99 | 3.36 | 4.91 | 6.48 | 9.98 | 20.26 | 30.30 |
| | | annual | 8039 | 6.31 | 4.25 | 0.20 | 2.55 | 3.68 | 5.00 | 7.59 | 14.21 | 57.81 |
| | | Spring | 2128 | 6.74 | 3.68 | 2.11 | 3.15 | 4.23 | 5.71 | 8.06 | 13.80 | 33.92 |
| | 2011 | Summer | 1903 | 6.14 | 4.58 | 0.20 | 1.91 | 3.51 | 4.84 | 7.29 | 13.77 | 38.45 |
| | | Fall | 1940 | 5.73 | 4.83 | 1.18 | 2.56 | 3.39 | 4.21 | 5.83 | 14.86 | 57.81 |
| | | Winter | 2068 | 6.58 | 3.81 | 1.69 | 2.49 | 3.79 | 5.36 | 8.40 | 14.39 | 25.12 |
| | | annual | 4459 | 5.70 | 3.79 | 0.38 | 2.12 | 3.48 | 4.56 | 6.75 | 13.26 | 50.32 |
| | | Spring | 192 | | | | | | | | | |
| OC | 2012 | Summer | 1259 | 5.19 | 3.00 | 1.74 | 2.64 | 3.39 | 4.11 | 5.66 | 11.54 | 21.87 |
| | | Fall | 1520 | 6.14 | 4.97 | 0.38 | 0.74 | 3.33 | 4.58 | 7.60 | 15.71 | 50.32 |
| | | Winter | 1488 | 5.81 | 2.94 | 1.76 | 2.68 | 3.90 | 4.99 | 6.99 | 11.29 | 30.17 |
| | | annual | 5547 | 8.93 | 6.18 | 0.36 | 3.08 | 4.88 | 6.79 | 11.39 | 20.88 | 62.05 |
| | | Spring | 1080 | 7.31 | 4.97 | 0.36 | 2.40 | 4.02 | 5.61 | 9.41 | 18.47 | 31.32 |
| | 2013 | Summer | 1570 | 9.27 | 5.04 | 1.11 | 3.87 | 5.51 | 7.82 | 12.05 | 18.75 | 39.66 |
| | | Fall | 1465 | 8.40 | 6.32 | 2.49 | 3.76 | 4.89 | 6.02 | 9.28 | 20.57 | 62.05 |
| | | Winter | 1432 | 10.33 | 7.50 | 2.03 | 2.82 | 5.14 | 7.86 | 13.29 | 26.28 | 49.03 |
| | | annual | 6914 | 7.83 | 4.55 | 0.81 | 3.66 | 4.95 | 6.52 | 9.09 | 17.01 | 47.10 |
| | | Spring | 1284 | 8.69 | 4.52 | 1.16 | 3.78 | 5.62 | 7.86 | 10.16 | 17.50 | 41.53 |
| | 2014 | Summer | 1829 | 6.53 | 2.85 | 0.81 | 3.37 | 4.55 | 5.83 | 7.79 | 11.93 | 25.14 |
| | | Fall | 2144 | 6.67 | 3.23 | 0.85 | 3.70 | 4.72 | 5.77 | 7.49 | 12.89 | 34.98 |
| | | Winter | 1657 | 10.13 | 6.22 | 2.27 | 3.98 | 5.67 | 8.19 | 12.82 | 22.39 | 47.10 |
| | | annual | 3531 | 3.41 | 2.07 | 0.62 | 1.34 | 2.01 | 2.73 | 4.08 | 8.27 | 20.46 |
| | | Summer | 1050 | 3.80 | 2.37 | 0.62 | 1.16 | 2.08 | 3.03 | 4.94 | 9.07 | 10.82 |
| OC/EC | 2010 | Fall | 1846 | 3.40 | 1.90 | 0.66 | 1.38 | 2.18 | 2.86 | 3.95 | 7.86 | 11.39 |
| | | Winter | 635 | 2.80 | 1.88 | 0.98 | 1.47 | 1.77 | 2.08 | 2.81 | 6.96 | 20.46 |
| | 2011 | annual | 8039 | 3.16 | 1.63 | 0.11 | 1.50 | 2.10 | 2.67 | 3.62 | 6.76 | 12.43 |

| Year | Season | N | | | | | | | | | |
|---|---|---|---|---|---|---|---|---|---|---|---|
| | Spring | 2128 | 3.19 | 1.43 | 0.89 | 1.71 | 2.30 | 2.82 | 3.54 | 6.35 | 10.08 |
| | Summer | 1903 | 3.19 | 1.66 | 0.11 | 1.04 | 2.11 | 2.88 | 3.88 | 6.59 | 12.43 |
| | Fall | 1940 | 3.29 | 1.72 | 1.06 | 1.60 | 2.11 | 2.70 | 3.85 | 7.05 | 10.99 |
| | Winter | 2068 | 2.98 | 1.71 | 0.92 | 1.51 | 1.91 | 2.40 | 3.17 | 7.02 | 9.98 |
| | annual | 4459 | 3.53 | 2.04 | 0.34 | 1.73 | 2.23 | 2.89 | 4.06 | 7.95 | 21.67 |
| | Spring | 192 | | | | | | | | | |
| 2012 | Summer | 1259 | 3.88 | 1.99 | 0.34 | 1.84 | 2.75 | 3.47 | 4.42 | 7.39 | 20.34 |
| | Fall | 1520 | 2.97 | 1.26 | 0.82 | 1.65 | 2.21 | 2.69 | 3.42 | 5.13 | 15.88 |
| | Winter | 1488 | 3.62 | 2.36 | 0.98 | 1.75 | 2.08 | 2.50 | 4.42 | 8.85 | 15.46 |
| | annual | 5547 | 3.92 | 1.76 | 0.46 | 2.05 | 2.83 | 3.57 | 4.59 | 6.71 | 28.04 |
| | Spring | 1080 | 3.51 | 1.52 | 0.46 | 1.79 | 2.58 | 3.23 | 4.05 | 6.30 | 18.50 |
| 2013 | Summer | 1570 | 4.49 | 2.20 | 1.13 | 2.28 | 3.23 | 4.08 | 5.17 | 7.73 | 28.04 |
| | Fall | 1465 | 4.02 | 1.58 | 1.11 | 2.32 | 2.99 | 3.72 | 4.64 | 6.67 | 19.51 |
| | Winter | 1432 | 3.49 | 1.30 | 1.13 | 2.00 | 2.60 | 3.25 | 4.11 | 5.81 | 13.84 |
| | annual | 6914 | 4.40 | 1.90 | 0.70 | 2.33 | 3.17 | 4.00 | 5.16 | 7.83 | 26.14 |
| | Spring | 1284 | 4.76 | 2.30 | 1.05 | 2.46 | 3.27 | 4.24 | 5.69 | 8.50 | 26.14 |
| 2014 | Summer | 1829 | 4.55 | 1.90 | 0.96 | 2.31 | 3.35 | 4.16 | 5.28 | 8.25 | 18.54 |
| | Fall | 2144 | 4.51 | 1.79 | 0.70 | 2.40 | 3.33 | 4.23 | 5.33 | 7.59 | 24.01 |
| | Winter | 1657 | 3.83 | 1.52 | 1.03 | 2.25 | 2.89 | 3.52 | 4.35 | 6.41 | 23.72 |

