# Peer review of "Assessment of carbonaceous aerosols in Shanghai, China, Part 1: Long-term evolution, seasonal variations and meteorological effects"

_Atmospheric Chemistry and Physics, 2017_

## Referee Comment (RC1) · Anonymous Referee #2 · 28 Apr 2017

This paper by Chang et al. presented a detailed analysis of five-year (2010-2015) on-line measurements of carbonaceous aerosols (CA) at an urban supersite in Shanghai, China. Temporal variations of organic carbon (OC) and elemental carbon (EC) concentrations are thoroughly explored. Then, they discussed the properties of OC and EC as a function of the meteorological conditions and the air mass origin. Moreover, the authors integrated the results with historical filter-based CA measurements and satellite-based AOD observations, concluding that ambient CA concentration in Shanghai has decreased since 2006 after the introduction of cleaner natural gas and the control of vehicular emissions. The large data set presented in the MS are unique and important in terms of aiding the validation of atmospheric chemistry modeling and informing the

effectiveness of air cleaning action to the public and policy-makers. Overall, this paper is well written and clearly describes the analysis, which addresses relevant scientific questions within the scope of ACP. I recommend this manuscript to be published after the following specific comments are addressed. (1) Title: the authors suggest that they are going to estimate secondary organic aerosol in their upcoming work. Given that the data used in their two MS are essentially the same, I suggest that the title of the current MS can be revised as "Assessment of carbonaceous aerosols in Shanghai, China. Part 1: Long-term evolution, seasonal variations and meteorological effects". (2) Page 7, lines 3-7: "concentrations of black carbon were continuously measured using an Aethalometer AE-31...880 nm wavelength is considered as the standard channel to determine BC concentrations". What is the mass absorption coefficient used for converting absorption coefficient to mass concentrations? (3) Page 10, line 25: I didn't see the citation. The difference between Sunset EC and Aethalometer BC is due to different techniques: Aethalometer is solely based on the optical technique while Sunset use the thermal-optical technique. Please note the paper by Petzold, A. et al. (2013), recommendations for reporting "black carbon" measurements, Atmos. Chem. Phys., 13(16), 8365-8379, doi: 10.5194/acp-13-8365-2013. (4) Page 16, lines 25-26: what is the specific chemical reactions? (5) Conclusion: new scientific findings need more emphasis in comparison with previous work. (6) Table 1: this is a big table of data and difficult to take in. Could it be simplified by removing some variables?

———————————————

---

## Referee Comment (RC2) · Anonymous Referee #1 · 11 May 2017

This manuscript presents a statistical analysis of long-term measurements of carbonaceous aerosols in Shanghai, China. Temporal and spatial variations of OC and EC concentration were investigated according to the observation results with reasonable scientific discussions and explanations. I recommend this article to be published after these minor comments as follows are answered properly.

Page 1-2: Please consider rephrasing the abstract. There are too many detailed discussions and general introductions, which make it not convenient to draw the key points from the current study. Please refer your conclusions.

Page 3, line 1-13: Please give proper references.

[Figure]

Page 3, line 28, Page 4, line 1-2, Page 4, line 3-15: What is the scientific information behind this? Please consider the scientific significance or overview relating to carbonaceous aerosols, no matter the study in China or abroad.

Page 5, line 1-7: Was there short-term analysis of OC/EC measurements obtained from the semi-continuous OC/EC analyzer? What were their findings or the general comparison with your study?

Page 5, line 14: When and where, please specify.

Page 5, line 24-25: Are these emission data for Shanghai or YRD, please specify.

Page 5, line 27: 'urban road network' or 'urban traffic network'?

Page 6, line 8: Please add proper references.

Page 6, line 15: There are actually four OC contents according to four different heating temperatures; have you ever analyzed them separately?

Page 7, line 7: Please add proper references.

Page 7, line 8-10: Please give specific names of all instruments, which measured CO, NO2 and SO2 concentration.

Page 8, line 8: Please consider rephrasing the sentence.

Page 8, line 9-10: Please give brief explanations why BPP and PSCF are better than back trajectory analysis in your case.

Page 10, line 5-6: Please rephrase the sentence.

Page 10, line 14-16: Please give references.

Page 10, line 24-25: How is BC measured differently than EC? Please specify. BC is measured based on optical properties, which is light-absorptive EC, hence, it should be lower than your measured EC, please give detail explanation, for instance, did they measure light absorbance using different wavelength lase?. And which earlier studies

are consistent with your results? Please list them.

Page 11, line 1-2: what kind of different thermal, optical and chemical behavior during springtime as you expect?

Page 11, line 9: Your PM2.5 measurement was only started from 2013, so please reword your sentence.

Page 11, line 10-12: This is a scientific article; it is better not only report data. Please give short or brief discussion, even though you give extensive discussions elsewhere.

Page 11, line 15: 56% and 23% of what?

Page 11, line 17-19: Please consider rewording your sentence.

Page 11, line 20: Should primary inorganic aerosol also contribute?

Page 11-12, line 25-end: In a scientific article, it is better not give such massive information and description of the local policies. And I doubt you can conclude that these policies have been ineffective in this study according to your simple observation data.

Page 12, line 8: What does your OC frequency mean in Fig. 4? Please specify in your method part.

Page 12, line 11: Please add references for the discussions on the most severely polluted month.

Page 12, line 11-13: I don't understand your statement here, please reword your sentence and give appropriate explanation.

Page 12, line 19: You concluded previously that the pollution control policies are ineffective; however, here you said the air-cleaning measures are successful. Please be consistent.

Page 12, line 23-25: Please give proper references here.

Page 12, line 29: How was it validated by the evolution of SO2 concentration, please

specify.

Page 13, line 12: It is really difficult to read Fig. 6, especially each subplot. Please consider replot them. For monthly and seasonal variations, I would suggest a figure plotting the average or medium concentrations of each month or each season during all these years. Then give further discussions regarding to the new figure.

Page 13, line 26-27: I would like to see the comparison of the concentration of OC from different seasons but of the same year.

Page 14, line 19-22: I don't get it. Please reword it.

Page 14, line 25, or Page 35, 37: Please make your figure caption more clearly. Which one is for weekdays and which one is for weekend?

Page 14 line 25 to Page 15, line 9: I suggest you plot the mass fraction of EC and OC in PM2.5 or PM10 first and then give the conclusions.

Page 15, line 17 to 24: This paragraph has nothing to do with the explanation of the diurnal patterns of EC and your CO emissions. Or at least it is not written in a clear way to explain your scientific issues here, please consider rewrite it.

Page 15, line 25: The scatter plots of EC vs. CO in Fig. 9 can not confirm that on-road traffic is an important source contributing to EC emissions in Shanghai. You need to cite similar works from others to support your conclusions here.

Page 15, line 27: Where does it show the multi-day build-up of OC for all months in your Fig. 3. Sorry I cannot capture it. Please specify.

Page 15, line 28: I have huge difficulties to read your figures in your manuscript, as the order of your figures are completely messed up. Please consider reorder all your figures in your manuscript.

Page 16, line 5: The increase as you stated could only be observed for the data during fall of 2010 and 2012. As I seen, your diurnal patterns of the OC con vary by seasons

and also by years. The complexity of the diurnal variation of OC, however, suggests the OC you measured were from different sources, both primary and secondary. It is really difficult to estimate only one OC emission source from your current data.

Page 16, line 14: This is actually quite interesting result from Fig. 11a. (Here, again, how could you talked about Fig.11b first, then Fig. 11a? It is difficult to follow.) There is obvious correlation, but not linear relationship. It will be more interesting if you could find out the changes of PM loadings or other trace gaseous concentration at O3>60 and O3<60, which might support what stated in the following part at current section.

Page 16, line 25: I don't fully agree with you. Higher T does not necessarily mean stronger solar radiation intensity. It also depends on your cloudiness. And at higher T, normally your OC evaporates more, which could not explain what you observed here.

Page 17, line 1-5: Similarly, at low T, the evaporation of OC is slow that it is reasonable to observe a higher OC concentration at low T. However, OC/EC is less than 2, which means the emissions of EC was quite high at those conditions, which you discussed in the next section. Please always combine OC and EC together when you discuss OC/EC ratio.

Page 17, line 6-11: Could that be your primary emissions of EC is high at low T, or winter time, as suggested by your Fig. 9. Please refer the absolute value of EC and CO. And why the WS-dependence of EC concentration is not valid for OC?

Page 17, line 13-20: Please add the correlation coefficient of OC or EC vs. WS.

Page 18, line 5-6: What is your expected reason?

Page 18, line 15: Which urban districts of Shanghai? Should the air mass from SW be your major source for EC?

Page 18, line 19: I am not sure this is fully correct. It is more depending on your back trajectory of the air mass, but not winds from Siberia, as it is quite clean background.

Page 18, line 24 to Page 19: What does rice-growing areas and biomass burning activity have any relationship with your PSCF or your current section (meteorological effect)? In which area, does the biomass burning occur normally in China, which contributes the EC concentration in Shanghai?

Page 19, line 11: What is the anthropogenic influence you stated here? Please specify. Should biomass burning belong to anthropogenic activity?

Page 19, line 15-line 18: Please consider removing it and giving your scientific understanding of the PSCF you plotted in Fig. 15.

---

## Author Comment (AC1) · 11 Jul 2017

The comment was uploaded in the form of a supplement:
https://www.atmos-chem-phys-discuss.net/acp-2017-50/acp-2017-50-AC1-supplement.pdf

---

## Author Comment (AC2) · 11 Jul 2017

We thank two referees for their careful considerations of the manuscript and their well thoughtful comments. These certainly helped to significantly improve the paper. We've addressed all comments and questions raised, and our point-by-point responses are shown below. The referees' comments are in *Italic* and our responses are in normal font. The revised manuscript is followed the response.

**Referee 1:**

*This paper by Chang et al. presented a detailed analysis of five-year (2010-2015) online measurements of carbonaceous aerosols (CA) at an urban supersite in Shanghai, China. Temporal variations of organic carbon (OC) and elemental carbon (EC) concentrations are thoroughly explored. Then, they discussed the properties of OC and EC as a function of the meteorological conditions and the air mass origin. Moreover, the authors integrated the results with historical filter-based CA measurements and satellite-based AOD observations, concluding that ambient CA concentration in Shanghai has decreased since 2006 after the introduction of cleaner natural gas and the control of vehicular emissions. The large data set presented in the MS are unique and important in terms of aiding the validation of atmospheric chemistry modeling and informing the effectiveness of air cleaning action to the public and policy-makers. Overall, this paper is well written and clearly describes the analysis, which addresses relevant scientific questions within the scope of ACP. I recommend this manuscript to be published after the following specific comments are addressed.*

Thanks for the favorable comments and the recognition of our work.

*(1) Title: the authors suggest that they are going to estimate secondary organic aerosol in their upcoming work. Given that the data used in their two MS are essentially the same, I suggest that the title of the current MS can be revised as "Assessment of*

*carbonaceous aerosols in Shanghai, China. Part 1: Long-term evolution, seasonal variations and meteorological effects".*

We agree and has been revised in the MS.

*(2) Page 7, lines 3-7: "concentrations of black carbon were continuously measured using an Aethalometer AE-31: 880 nm wavelength is considered as the standard channel to determine BC concentrations". What is the mass absorption coefficient used for converting absorption coefficient to mass concentrations?*

The manufacturer's recommended value for $\alpha_{ap}$ is $14625/\lambda$, which is based upon calibrations during instrument development and theoretical calculations. It has a value of 16.6 m$^2$ g$^{-1}$ at $\lambda$=880 nm. This accounts for absorption by BC and additional light attenuation assumed to be caused by multiple scattering within the filter media.

*(3) Page 10, line 25: I didn't see the citation. The difference between Sunset EC and Aethalometer BC is due to different techniques: Aethalometer is solely based on the optical technique while Sunset use the thermal-optical technique. Please note the paper by Petzold, A. et al. (2013), recommendations for reporting "black carbon" measurements, Atmos. Chem. Phys., 13(16), 8365-8379, doi: 10.5194/acp-13-8365-2013.*

Thanks for the recommendation of this reference. We've added a new reference in the revised MS (Venkatachari et al., 2006).

Reference:

Venkatachari, P., Zhou, L., Hopke, P. K., Schwab, J. J., Demerjian, K. L., Weimer, S., Hogrefe, O, Felton, D., and Rattigan, O.: An intercomparison of measurement

methods for carbonaceous aerosol in the ambient air in New York City, Aerosol Sci. Technol., 40(10), 788-795, doi: 10.1080/02786820500380289, 2006.

*(4) Page 16, lines 25-26: what is the specific chemical reactions?*

We've followed the suggestion given by Referee 2 to delete this sentence in the revised MS.

*(5) Conclusion: new scientific findings need more emphasis in comparison with previous work.*

This work represents the first multi-year and near real-time measurement study of carbonaceous aerosols in $PM_{2.5}$ in China. We think the most important finding is that carbonaceous aerosol pollution in Shanghai has gradually reduced since 2006, which has not been reported before. Our results confirm the success of replacing coal with cleaner energy such as natural gas in Shanghai, which can be adopted in other megacities like Beijing and Guangzhou to curb $PM_{2.5}$ pollution.

*(5) Table 1: this is a big table of data and difficult to take in. Could it be simplified by removing some variables?*

We agree and have deleted the variables of "$OC/PM_{2.5}$" and "$EC/PM_{2.5}$" in the revised MS.

**Referee 2:**

*This manuscript presents a statistical analysis of long-term measurements of carbonaceous aerosols in Shanghai, China. Temporal and spatial variations of OC and EC concentration were investigated according to the observation results with reasonable scientific discussions and explanations. I recommend this article to be published after these minor comments as follows are answered properly.*

Many thanks for the encouraging comments and useful suggestions. After looking through the comments, we noticed that there were lots of minor mistakes in our original MS which can be avoided before submission. We are grateful for such generous contribution of your expertise and valuable time.

*Page 1-2: Please consider rephrasing the abstract. There are too many detailed discussions and general introductions, which make it not convenient to draw the key points from the current study. Please refer your conclusions.*

We agree that the current abstract is some sort of tediously long. We've revised the abstract as below (totally 364 words):

Carbonaceous aerosols are major chemical components of fine particulate matter ($PM_{2.5}$) with major impacts on air quality, climate change, and human health. Gateway to fast-rising China and home of over twenty million people, Shanghai throbs as the nation's largest mega city and the biggest industrial hub. From July 2010 to December 2014, hourly mass concentrations of ambient organic carbon (OC) and elemental carbon (EC) in the $PM_{2.5}$ fraction were quasi-continuously measured in Shanghai's urban center. The annual OC and EC concentrations (mean $\pm$ 1 $\sigma$) in 2013 (8.9$\pm$6.2 and 2.6$\pm$2.1 µg m$^{-3}$, n=5547) and 2014 (7.8$\pm$4.6 and 2.1$\pm$1.6 µg m$^{-3}$, n=6914) were higher than that of 2011 (6.3$\pm$4.2 and 2.4$\pm$1.8 µg m$^{-3}$, n=8039) and 2012 (5.7$\pm$3.8 and 2.0$\pm$1.6 µg m$^{-3}$, n=4459). We integrated the results from historical field measurements (1999-2012) and satellite

observations (2003-2013), concluding that carbonaceous aerosol pollution in Shanghai has gradually reduced since 2006. In terms of monthly variations, average OC and EC concentrations ranged from 4.0 to 15.5 and from 1.4 to 4.7 $\mu g\ m^{-3}$, accounting for 13.2-24.6% and 3.9-6.6% of the seasonal $PM_{2.5}$ mass (38.8-94.1 $\mu g\ m^{-3}$), respectively. The concentrations of EC (2.4, 2.0, 2.2, 3.0 $\mu g\ m^{-3}$ in spring, summer, fall, and winter, respectively) showed little seasonal variation (excepting winter) and weekend-weekday dependence, indicating EC are a relatively stable constitute of $PM_{2.5}$ in the Shanghai urban atmosphere. In contrast to OC (7.3, 6.8, 6.7, and 8.1 $\mu g\ m^{-3}$ in spring, summer, fall, and winter, respectively), EC showed marked diurnal cycles and correlated strongly with CO across all seasons, confirming vehicular emissions as the dominant source of EC at the targeted site. Our data also reveal that both OC and EC showed concentration gradients as a function of wind direction and wind speed, generally with higher values associated with winds from the southwest, west, and northwest. This was consistent with their higher potential as source areas, as determined by the potential source contribution function analysis. A common high potential source area, located along the middle and lower reaches of the Yangtze River instead of Northern China, was pinpointed during all seasons. These results demonstrate that the measured carbonaceous aerosols were driven by the interplay of local emissions and regional transport.

*Page 3, line 1-13: Please give proper references.*

Three relevant references have been inserted in the revised MS.

Turpin, B. J., and Huntzicker, J. J.: Secondary formation of organic aerosol in the Los Angeles basin: A descriptive analysis of organic and elemental carbon concentrations, Atmos. Environ., 25, 207-215, doi: 10.1016/0960-1686(91)90291-E, 1991.

Bond, T. C., Doherty, S. J., Fahey, D., Forster, P., Berntsen, T., DeAngelo, B., Flanner, M., Ghan, S., Kärcher, B., and Koch, D.: Bounding the role of black carbon in the climate system: A scientific assessment, J. Geophys. Res., 118, 5380-5552, doi: 10.1002/jgrd.50171, 2013.

5  Hallquist, M., Wenger, J. C., Baltensperger, U., Rudich, Y., Simpson, D., Claeys, M., Dommen, J., Donahue, N. M., George, C., Goldstein, A. H., Hamilton, J. F., Herrmann, H., Hoffmann, T., Iinuma, Y., Jang, M., Jenkin, M. E., Jimenez, J. L., Kiendler-Scharr, A., Maenhaut, W., McFiggans, G., Mentel, T. F., Monod, A., Prévôt, A. S. H., Seinfeld, J. H., Surratt, J. D., Szmigielski, R., and Wildt, J.: The
10  formation, properties and impact of secondary organic aerosol: current and emerging issues, Atmos. Chem. Phys., 9, 5155-5236, doi: 10.5194/acp-9-5155-2009, 2009.

*Page 3, line 28, Page 4, line 1-2, Page 4, line 3-15: What is the scientific information*

15  *behind this? Please consider the scientific significance or overview relating to carbonaceous aerosols, no matter the study in China or abroad.*

We are sorry for making the reviewer confused and this sentence has been deleted in the revised MS.

20  *Page 5, line 1-7: Was there short-term analysis of OC/EC measurements obtained from the semi-continuous OC/EC analyzer? What were their findings or the general comparison with your study?*

We appreciate the constructive comment. There are several papers (e.g., Cheng et al., Environ. Sci. Technol. 2015, 49, 831-838) regarding the short-term analysis of OC/EC
25  measurements obtained from the semi-continuous OC/EC analyser in China. However,

in previous work, OC and EC were typically discussed with other PM$_{2.5}$ chemical components together, and it was difficult to tease out the characteristics of OC and EC. Moreover, previous work focused on the short-term (e.g., several hours to several days) formation mechanism of haze pollution, thus it makes little sense to perform a comparison with our results. The current MS is mainly about the investigation of temporal evolution of OC and EC mass concentrations. As another reviewer mentioned, there will be a companion work regarding the formation of secondary organic aerosols, in which our findings will have more opportunities to compare with previous work.

*Page 5, line 14: When and where, please specify.*

We've specified the sampling period (from 10 June 2010 to 31 December 2014) and location (Shanghai) in the revised MS.

*Page 5, line 24-25: Are these emission data for Shanghai or YRD, please specify.*

Specified as Shanghai.

*Page 5, line 27: 'urban road network' or 'urban traffic network'?*

Revised as "urban traffic network"

*Page 6, line 8: Please add proper references.*

Two references were added.

Turpin, B. J., Saxena, P., Andrews, E., 2000. Measuring and simulating particulate organics in the atmosphere: problems and prospects. Atmos. Environ. 34, 2983-3013, 2000.

NIOSH: Elemental Carbon (Diesel Particulate): Method 5040, in: NIOSH Manual of Analytical Methods, 4th edition, edited by: Eller, P. M. and Cassinelli, M. E., National Institute for Occupational Safety and Health, DHHS (NIOSH), Cincinnati, OH, USA, Publication No. 96-135, 1996.

*Page 6, line 15: There are actually four OC contents according to four different heating temperatures; have you ever analyzed them separately?*

Although the separation and discussion of OC contents is beyond the scope of this work, we think this is a useful comment and will certainly to be taken into account in our future sampling work. Given the highly complex nature of OC, we are going to deploy an aerosol time-of-flight mass spectrometer to provide real-time information of OC contents.

*Page 7, line 7: Please add proper references.*

We've added a reference in the revised MS:

Arnott, W. P., Hamasha, K., W. Moosmüller, H., Sheridan, P. J., and Ogren, J. A., Towards aerosol light-absorption measurements with a 7-wavelength Aethalometer: Evaluation with a photoacoustic instrument and 3-wavelength Nephelometer, Aerosol Sci. Technol., 39, 17-29, doi: 10.1080/027868290901972, 2005.

*Page 7, line 8-10: Please give specific names of all instruments, which measured CO,*

*$NO_2$ and $SO_2$ concentration.*

According to the information provided by Shanghai Pudong Environmental Monitoring Center, ambient CO, $NO_2$ and $SO_2$ concentrations were measured by Thermo 48i, 42i, and 43i, respectively. We've added them in the revised MS.

*Page 8, line 8: Please consider rephrasing the sentence.*

We've revised the sentence as "Back trajectories analysis is a commonly used technique in atmospheric sciences, while at the city scale, back trajectories make little sense" because it was generally based on wild guess and illogical reasoning.

*Page 8, line 9-10: Please give brief explanations why BPP and PSCF are better than*

*back trajectory analysis in your case.*

The wind speed (WS) and wind direction (WD) in BPP are measured at near ground level (18 m a.g.l), while the WS and WD in PSCF are calculated at a much higher height (typically 500 m a.g.l). Therefore, BPP is more suitable for tracing the origins of air masses at city scale.

*Page 10, line 5-6: Please rephrase the sentence.*

This sentence has been revised as "Between 10 June 2010 and 31 December 2014, the Sunset carbon analyser was successfully operated during 75% of the time".

*Page 10, line 14-16: Please give references.*

There are various technical definitions of four seasons, but local usage of the term varies according to local climate, cultures and customs. In most Northern Hemisphere temperate locations, spring months are March, April and May (summer is June, July, August; autumn is September, October, November; winter is December, January, February) (see http://glossary.ametsoc.org/wiki/Spring), although differences exist from country to country. There are several versions of "four seasons" in China. Seasons often held special significance for rural citizens in China, whose lives revolved around planting and harvest times, and the change of seasons was often attended by ritual. Nevertheless, all versions of dividing a season are very close to each other. There is no clear classification criterion to divide four seasons for investing the characteristics of air pollutants in China. Here we adopted an unspoken or default rule to divide four seasons in Shanghai. This rule has been adopted in previous studies (e.g., Sun et al., 2015).

Reference:

Sun et al., Long-term real-time measurements of aerosol particle composition in Beijing, China. Atmos. Chem. Phys., 15, 10149-10165, 2015

*Page 10, line 24-25: How is BC measured differently than EC? Please specify. BC is measured based on optical properties, which is light-absorptive EC, hence, it should be lower than your measured EC, please give detail explanation, for instance, did they measure light absorbance using different wavelength lase? And which earlier studies are consistent with your results? Please list them.*

Thanks for this thought-provoking comment. EC is operationally defined by their analysis method or protocol (e.g., IMPROVE/TOR, NIOSH/TOT). When the components of the ambient carbonaceous aerosol are considered thermally different,

the EC fraction is called thermal EC, and is usually determined by TOT or TOR analysis. When carbonaceous aerosol components are considered optically different, the strongly light absorbing fraction is known as BC. BC can be measured using a variety of optical absorption methods, and the Aethalometer is one of the most commonly utilized, and easy-to-use techniques employed to measure real-time BC concentrations. BC is generally produced during incomplete high-temperature combustion. It is affected to a limited extent by atmospheric processing and is therefore a direct tracer for combustion emissions. The various chemical and physical properties of the EC may cause changes in the measured optical BC concentrations even if the same measurement system is used.

The measurement of BC is complicated by the lack of a simple definition of BC and the absence of techniques that are uniquely sensitive to BC. We agree with the reviewer that BC is essentially light-absorptive EC. Practically, despite multiple wavelengths operated, the Aethalometer generally treats light-absorptive carbon as BC (equivalent black carbon), which may overestimate BC concentration because the presence of light-absorptive organic carbon, also known as brown carbon. The BC measurements used here are based on the 880-nm wavelength to minimize potential interference from brown carbon.

Although EC concentration is generally higher than BC concentration, our results are generally consistent with Venkatachari et al (2006).

[Figure]

We've reworded this sentence as "The highest degree of correlation ($R^2 = 0.71$) was observed during wintertime when the seasonal OC/EC ratio was the lowest (Table 1), suggesting that primary emissions are an important source contributing to both OC and EC in Shanghai during the cold season".

*Page 14, line 25, or Page 35, 37: Please make your figure caption more clearly. Which one is for weekdays and which one is for weekend?*

We've added the specific row and column in the text to make our description clear, e.g., "fall of 2012 (the third row, the second column)".

*Page 14 line 25 to Page 15, line 9: I suggest you plot the mass fraction of EC and OC in $PM_{2.5}$ or $PM_{10}$ first and then give the conclusions.*

We didn't have $PM_{10}$ data in the current study. Since October 2016, an 8-stages PM sampler was deployed at the Shanghai Pudong supersite. We look forward to showing the size-segregated EC and OC aerosols in the near future.

*Page 15, line 17 to 24: This paragraph has nothing to do with the explanation of the diurnal patterns of EC and your CO emissions. Or at least it is not written in a clear way to explain your scientific issues here, please consider rewrite it.*

Agree, and we've deleted this paragraph in the revised MS.

*Page 15, line 25: The scatter plots of EC vs. CO in Fig. 9 cannot confirm that on-road traffic is an important source contributing to EC emissions in Shanghai. You need to cite similar works from others to support your conclusions here.*

Agree, and we've cited two similar works in China to support our conclusion.

5    1.  Chen, D., Cui, H., Zhao, Y., Yin, L., Lu, Y., and Wang, Q.: A two-year study of carbonaceous aerosols in ambient $PM_{2.5}$ at a regional background site for western Yangtze River Delta, China, Atmos. Res., 183, 351-361, doi: 10.1016/j.atmosres.2016.09.004, 2017.

2.  Yang, F., He, K., Ye, B., Chen, X., Cha, L., Cadle, S. H., Chan, T., and Mulawa, P.
10   A.: One-year record of organic and elemental carbon in fine particles in downtown Beijing and Shanghai, Atmos. Chem. Phys., 5, 1449-1457, https://doi.org/10.5194/acp-5-1449-2005, 2005.

Still, CO variation can be utilized as a robust indicator of vehicle emissions in Shanghai. Historically, CO emissions in Shanghai and its surrounding YRD region mainly came
15   from iron and steel manufacturing and on-road vehicles, which contributed 34% and 30% of the total, respectively in 2007 (Huang et al., 2011). Due to changing economic activity, emission sources of air pollutants in China are changing rapidly. For example, over the past several years, China has implemented a portfolio of plans to phase out its old-fashioned and small steel mills, and raise standards for industrial pollutant
20   emissions (Chang et al., 2012). In contrast, China continuously experienced double digit growth in terms of auto sales during the same period, and became the world's largest automobile market since 2009 (Chang, 2014). Consequently, on-road traffic has overtaken industrial sources as the dominant source of CO emissions in Eastern China (Zhao et al., 2012). In our previous work in Shanghai (Chang et al., 2016), CO shows
25   a well-marked bimodal diurnal profile, with maxima in the morning (starting at 05:00

local time) and the evening (starting at 16:00), consistent with the variation of traffic flow in Shanghai (Liu et al., 2012).

We confess that estimated solely based on our current data, it is difficult to pinpoint a specific OC emission source. Therefore, we've added the potential limitation and soften our claim in the revised MS as "In this study, OC in fall and winter showed a clear daytime increase until late afternoon, illustrating possible role played by gas-phase photochemical processing in driving the OC diurnal cycle. However, it should be noticed that it is still difficult to pinpoint a specific OC emission source based on our current data. Since ambient OC concentration depends on multiple factors, further study will reveal more information about the different formation mechanisms involved".

*Page 16, line 14: This is actually quite interesting result from Fig. 11a. (Here, again, how could you talk about Fig.11b first, then Fig. 11a? It is difficult to follow.) There is obvious correlation, but not linear relationship. It will be more interesting if you could find out the changes of PM loadings or other trace gaseous concentration at O₃>60 and O₃<60, which might support what stated in the following part at current section.*

We've corrected the mess of the order of figures in the revised MS. Again, sorry for making the difficulty to follow.

We admire the reviewer for his/her shrewder observation. Indeed, setting $O_3=60$ as a breaking point, the linear correlation between $O_3$ and OC can be shifted from negative ($O_3<60$) to positive ($O_3>60$) as indicated in the figure below. In fact, this interesting result has been involved in our upcoming work regarding the formation of secondary organic aerosols in Shanghai.

[Figure]

*Page 16, line 25: I don't fully agree with you. Higher T does not necessarily mean stronger solar radiation intensity. It also depends on your cloudiness. And at higher T, normally your OC evaporates more, which could not explain what you observed here.*

We agree, and we've deleted this sentence in the revised MS.

*Page 17, line 1-5: Similarly, at low T, the evaporation of OC is slow that it is reasonable to observe a higher OC concentration at low T. However, OC/EC is less than 2, which means the emissions of EC was quite high at those conditions, which you discussed in the next section. Please always combine OC and EC together when you discuss OC/EC ratio.*

Thanks for the suggestion, and we've combined the discussion of OC/EC ratio with OC or EC in the revised MS. The current work is mainly focus on the discussion in terms of the mass concentrations and temporal variations of EC and OC, and a detailed discussion regarding secondary organic aerosols (including OC/EC ratio) will be presented in our next work.

*Page 17, line 6-11: Could that be your primary emissions of EC is high at low T, or winter time, as suggested by your Fig. 9. Please refer the absolute value of EC and CO. And why the WS-dependence of EC concentration is not valid for OC?*

On-road traffic and biomass burning (especially burning of crop residues) are two major sources of EC emissions in China, and biomass burning activities are concentrated in summer and generally rare during wintertime. As to traffic-emitted EC, there is no evidence to show a clear relationship between ambient temperature and emission amount. In Fig. 9, the absolute values of both EC and CO are much higher than other seasons, and this is not necessarily the result of higher emissions in winter but more likely due to a lower PBHL (Chang et al., 2016) and stronger regional transport during wintertime (discussed in latter section).

A greater dependence on wind speed for EC than OC is because that EC is primarily originated from local traffic emissions while OC is the complex product of primary emission (mainly local sources) and secondary formation (long-range transport and atmospheric ageing).

The primary reason is because of the different source regions between carbonaceous aerosols (Southwest Shanghai) and coal-related pollutants like sulfate (Northern China), which had been further discussed in the latter section.

*Page 18, line 15: Which urban districts of Shanghai? Should the air mass from SW be your major source for EC?*

Sorry for making this confusion. Yes, the urban districts of Shanghai are in its SW. We've revised as "Given the relatively low WS ($< 6$ m s$^{-1}$ in most cases; Fig. 14) in the

SW (where the man urban districts located), the urban areas of Shanghai were highly likely the major source areas for EC".

*Page 18, line 19: I am not sure this is fully correct. It is more depending on your back*

*trajectory of the air mass, but not winds from Siberia, as it is quite clean background.*

Shanghai has a humid subtropical climate and experiences four distinct seasons. Winters are chilly and damp, with northwesterly winds from Siberia can cause nighttime temperatures to drop below freezing, although most years there are only one or two days of snowfall. Despite the clean air in Siberia, winter air masses from Siberia

10  will mixed with extremely polluted air in Northern China (Huang et al., 2014).

We've revised as "with northwesterly winds from Siberia transporting high levels of air pollutants in Northern China to the YRD region resulting in poor air quality in Shanghai" in the revised MS to avoid potential misunderstanding.

20   *Page 19, line 11: What is the anthropogenic influence you stated here? Please specify.*

*Should biomass burning belong to anthropogenic activity?*

Thanks for the suggestion. We would like to clarify that we had specified the anthropogenic influence in lines 12-15, i.e., rapid urbanization process and massive industrial production. For biomass burning, pollutants' emissions from prescribed

25   burning (e.g., forest clearing, burning of crop residues) and fireplace/woodstove

activities are also anthropogenic origin, while these were not included in our discussion of "anthropogenic influence". Therefore, we've clarified this in revised MS.

[revised manuscript text omitted]